# Evolutionarily distinct lineages of a migratory bird of prey show divergent responses to climate change

Accurately predicting species' responses to anthropogenic climate change is hampered by limited knowledge of their spatiotemporal ecological and evolutionary dynamics. We combine landscape genomics, demographic reconstructions, and species distribution models to assess the eco-evolutionary responses to past climate fluctuations and to future climate of an Afro-Palaearctic migratory raptor, the lesser kestrel (*Falco naumanni*). We uncover two evolutionarily and ecologically distinct lineages (European and Asian), whose demographic history, evolutionary divergence, and historical distribution range were profoundly shaped by past climatic fluctuations. Using future climate projections, we find that the Asian lineage is at higher risk of range contraction, increased migration distance, climate maladaptation, and consequently greater extinction risk than the European lineage. Our results emphasise the importance of providing historical context as a baseline for understanding species' responses to contemporary climate change, and illustrate how incorporating intraspecific genetic variation improves the ecological realism of climate change vulnerability assessments.

Anthropogenic climate change is rapidly altering Earth's environmental conditions, exacerbating the effects of land use change and habitat loss on global biodiversity decline and further threatening the persistence of many species[1,2]. To track fast-changing environments, organisms can relocate to more suitable areas, respond via phenotypic plasticity, or undergo rapid adaptation[3–5]. However, not all populations or species can respond quickly enough, resulting in local or global declines, and even extinctions[6,7]. In this context, we argue that spatiotemporal variability in a species' ecology and evolutionary dynamics can be critical for its persistence in the face of climate change. However, most studies predicting species responses to climate change neglect intraspecific variation (e.g., differences in climate tolerances between populations) and lack historical context (*i.e.*, how the species has responded to climatic fluctuations in the past). Disregarding this information can lead to inaccurate predictions about a species' fate and to ineffective conservation efforts[8–10].

Migratory birds share many ecological features that make them highly suitable for investigating intraspecific sensitivity to climate change. Firstly, they may show remarkable inter-population variation in migratory strategies, whereby different populations rely on multiple geographical regions as they move between their breeding and non-breeding grounds, encompassing a wide variety of habitats and climatic zones[11,12], where they may face heterogeneous climatic changes[13]. Secondly, they are able to track seasonal changes in habitat availability[14] and develop novel migratory strategies and routes in response to changing seasonal resources, even over short time scales[15,16]. For example, adjustments in phenology and migration distance have been observed in response to changing climatic conditions[17–19]. Finally, while migratory birds are currently declining at a faster rate than non-migratory species[20–22], the ability of some species to track rapidly changing climates has been shown to partly buffer these declines[23–26].

The lesser kestrel (*Falco naumanni*) is a migratory bird of prey that breeds in open drylands of the Palaearctic region and moves to sub-Saharan Africa during the non-breeding season[27] (Fig. 1a). In the late 20th century, the species suffered a rapid demographic collapse

✉ e-mail: joan.ferrer.obiol@gmail.com; diego.rubolini@unimi.it

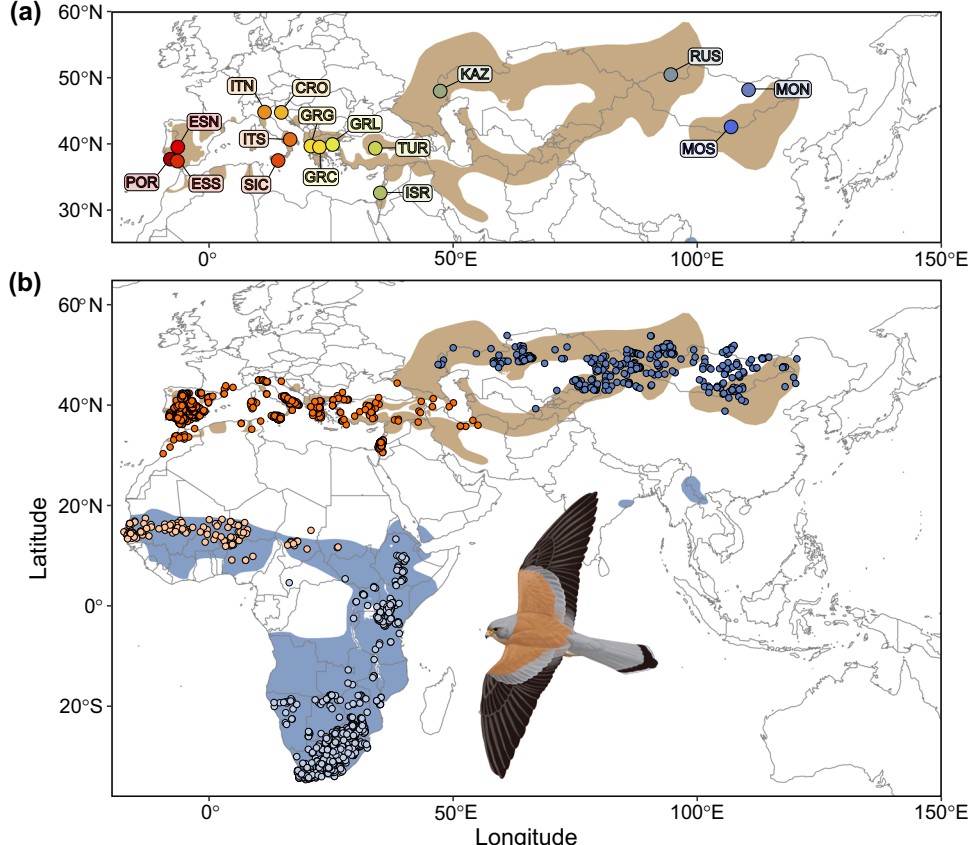

**Fig. 1 | Map showing sampling localities for lesser kestrel genetic data were obtained and the geographic distribution of occurrence records. a** Localities (coloured circles; $n = 16$) from which we obtained double-digest Restriction-Site Associated DNA (ddRAD) or mitogenome data; acronyms and sample size for each locality are reported in Supplementary Table 1. **b** Map of breeding (dark-shaded circles) and non-breeding (light-shaded circles) occurrence records (original occurrence data pooled for 2.5 arc-minute grid cells) used for species distribution modelling of Western (orange circles) and Eastern (blue circles) evolutionary significant units (ESUs). Lesser kestrel breeding and non-breeding distribution ranges are shown in brown and blue, respectively[30]. Background maps were obtained from the rnaturalearth v.0.3.2 R package. Lines delimiting countries are shown to facilitate map interpretation and do not necessarily represent accepted national boundaries. The lesser kestrel illustration (male) is used with permission from Martí Franch ©. Data underlying all components of Fig. 1 are provided at https://doi.org/10.5281/zenodo.14988067.

across its European range, linked to agricultural intensification and a prolonged drought in the Sahel[28]. Recent evidence suggests that European populations are responding to climate change by shifting northward both their breeding and non-breeding distribution ranges[29], yet whether other populations across its global distribution show similar responses is unknown[30]. Here, we provide an in-depth assessment of the vulnerability of the lesser kestrel to climate change by (1) inferring intraspecific evolutionary lineages, (2) assessing their ecological differentiation, and (3) investigating lineage-specific demographic, distributional and genomic responses to past and future climatic fluctuations across its global distribution range.

## Results

### Range-wide genetic differentiation and gene flow

To investigate range-wide patterns of genetic variation, we generated a chromosome-level reference genome (Supplementary Figs. 1 and 2; Supplementary Table 2) and genome-wide data for 119 individuals from 16 different localities (Fig. 1a; Supplementary Table 1, Supplementary Data 1), including 73,373 single nucleotide polymorphisms (SNPs) and complete mitogenomes. Genetic differentiation analyses coherently indicated the existence of two distinct lineages: a Western lineage including all the European and Middle Eastern populations and an Eastern lineage including all populations from Central and Eastern Asia (Fig. 2a, b; Supplementary Figs. 3, 4 and 5). Within the Western lineage, the individuals from Israel, and to a lesser extent those from

Turkey (and some from Greece and Italy), had some Eastern lineage ancestry, suggesting low levels of gene flow from the Eastern to the Western lineage. These two main clusters could be further subdivided into four finer-scale sub-clusters (Iberian peninsula, Italian and Balkan peninsulas, Israel, and Asia) (Fig. 2b; Supplementary Fig. 6). Phylogenetic analysis of mitogenomes identified three haplogroups (A, B and C; Fig. 2c, Supplementary Fig. 7), whose phylogenetic patterns were coherent with those derived from genome-wide SNPs. Using the Estimated Effective Migration Surfaces (EEMS) method, based on a stepping-stone dispersal model, we detected a main barrier to gene flow between the Western and Eastern lineages coinciding with the Caucasus mountains, and two additional barriers in Europe: one separating the small Croatian population, and another separating the Iberian populations (Fig. 2c). Consistent with low differentiation within each lineage, we detected higher gene flow than expected under isolation-by-distance (IBD) throughout Central and Eastern Asia, as well as between sub-clusters throughout the Mediterranean (Fig. 2c), and short within-lineage branch lengths in a maximum likelihood (ML) population tree (Fig. 2d, e). The only exception was the small and geographically isolated Croatian population[31], which also showed the lowest heterozygosity (Supplementary Fig. 8). Overall, levels of genome-wide heterozygosity were high for both Western and Eastern lineages (Supplementary Fig. 9). The combination of (1) lower genetic distances between the Eastern lineage and the outgroup compared to the Western lineage (as shown in the neighbour-net network), and

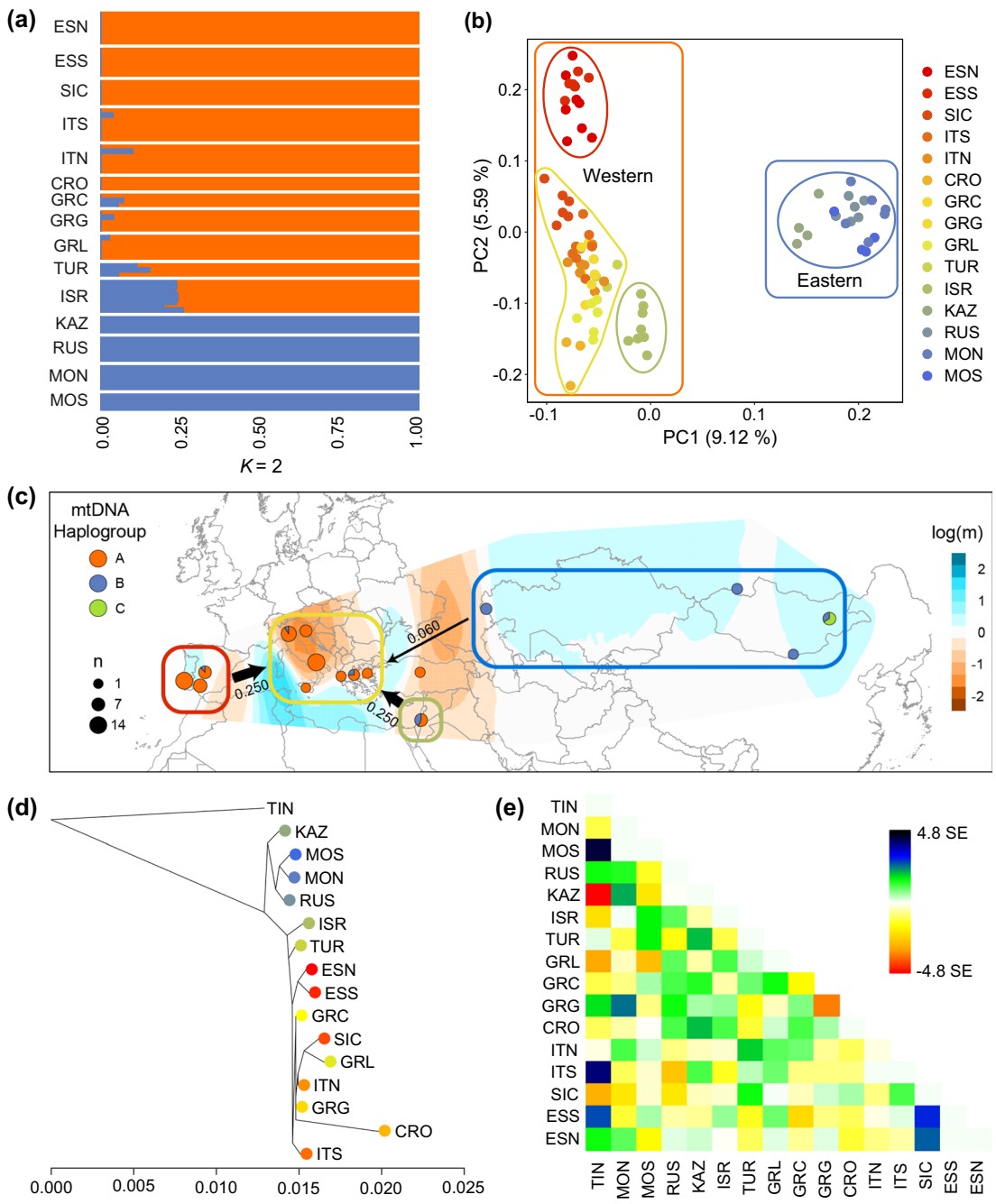

**Fig. 2 | Genetic differentiation and gene flow among lesser kestrel populations.** **a** Admixture component profiles for each sampling locality at $K = 2$ (best $K$ value based on cross-validation; Supplementary Fig. 11) based on double-digest Restriction-Site Associated DNA (ddRAD) data, showing the Western (orange) and Eastern evolutionarily significant units (ESUs) (blue). Supplementary Table 1 shows acronyms and details of sample size for each sampling locality. **b** Principal Component Analysis (PCA) based on 27,853 unlinked single-nucleotide polymorphisms (SNPs) highlighting the two ESUs (orange: Western ESU; blue: Eastern ESU) and the four fine-scale genetic clusters identified by fineRADStructure (Supplementary Fig. 6; coloured ellipses: Iberian peninsula in red, Italian and Balkan peninsulas - including Turkey - in yellow, Israel in green, Asia in blue). **c** EEMS-predicted barriers to gene flow (orange) showing a main barrier between the Western and Eastern ESUs. Pie charts show the frequencies of the three major mitochondrial haplogroups for each sampling locality ($n = 89$ individuals). For each of the fine-scale genetic clusters (coloured boxes; colours defined in **b**), the

arrows show the predicted fraction of immigrating individuals per generation (proportional to the size of the arrow; numbers shown close to each arrow) from other clusters (prediction based on *BayesAss3-SNPs* analysis). **d** Maximum-likelihood population tree inferred in Treemix using *Falco tinnunculus* (TIN) as an outgroup. **e** Residuals of the observed versus predicted squared allele frequency difference inferred in Treemix, expressed as the standard error of the deviation. Residuals above zero represent populations that share more genetic variation than predicted by the best-fit tree, potentially due to gene flow or shared ancestral genetic variation. Negative residuals represent populations that share less genetic variation than predicted by the best-fit tree. Data underlying all components of Fig. 2 are provided at https://doi.org/10.5281/zenodo.14988067. Background maps were obtained from the rnaturalearth v.0.3.2 R package. Lines delimiting countries are shown to facilitate map interpretation and do not necessarily represent accepted national boundaries.

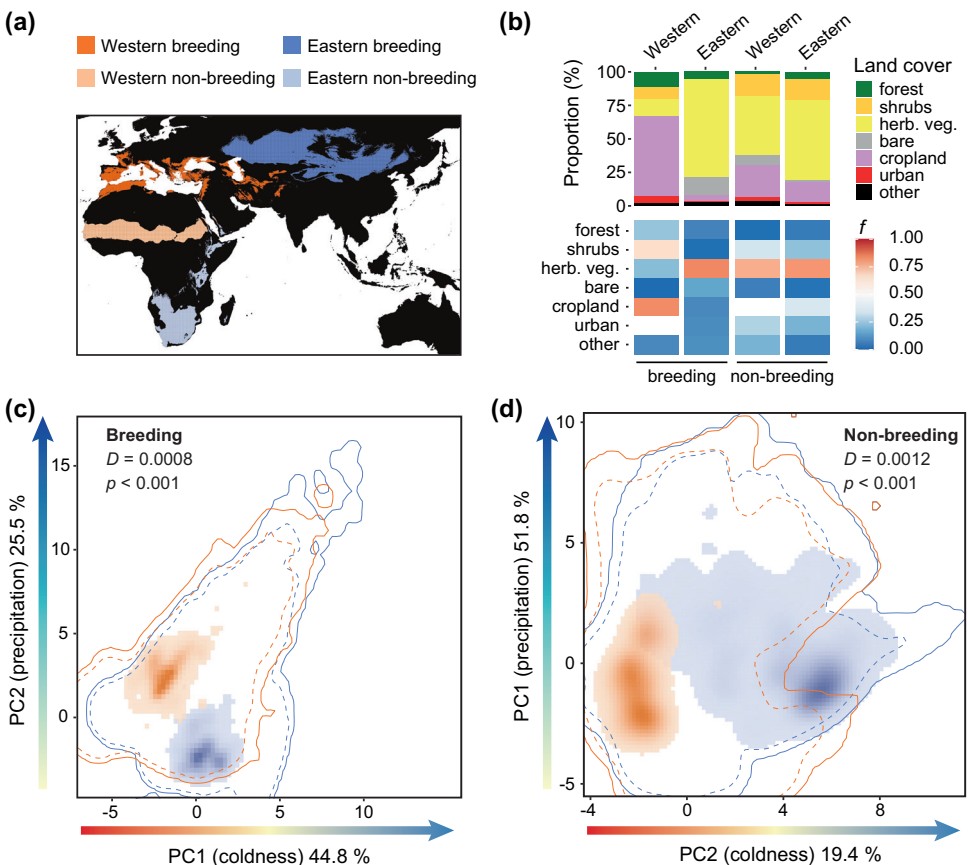

**Fig. 3 | Ecological differentiation between Western and Eastern lineages of the lesser kestrel. a** Reconstructed breeding and non-breeding ranges for Western and Eastern evolutionarily significant units (ESUs) obtained through species distribution models based on selected bioclimatic variables, largely matching the known distribution ranges of the species (Fig. 1, Supplementary Fig. 14). **b** (Top) Habitat use for each combination of lineage and season calculated as the proportion of seven main land cover categories (pooled from 22 original land use categories from the Copernicus Land Monitoring Service; Supplementary Table 3) in all 2.5 arc-minute grid cells with breeding or non-breeding occurrence records. (Bottom) Heatmap of habitat preference/avoidance for each of the seven main land cover categories based on a sign test of habitat selection. White cells ($f$ - 0.5) indicate habitats used proportionally to their availability, red cells ($f > 0.5$) indicate preferred habitats and blue cells ($f < 0.5$) indicate avoided habitats. **c**–**d** Climatic niche comparison between Western and Eastern ESUs in a two-dimensional space defined by the first two axes of a principal component analysis (PCA) of available climatic conditions across breeding (**c**) and non-breeding (**d**) ranges of the Western (orange contour lines) and Eastern (blue contour lines) lineages. The x-axes represent a gradient of increasing coldness (warmer climates in darker red, colder climates in darker blue), whereas the y-axes represent a gradient of increasing precipitation (beige to darker blue). The solid and dashed contour lines represent 100% and 75% of the available (background) climate, respectively. Coloured areas represent climatic niches (kernel densities of the climatic conditions at occurrence records) of Western (orange) and Eastern (blue) ESUs, with darker colours denoting higher densities and transparency adjusted to facilitate the visualisation of overlaps. Schoener's D index of niche overlap (0 = no overlap, 1 = full overlap) and the $p$-values of niche equivalency tests performed with the ecospat.niche.equivalency.test function from the ecospat R package (option overlap.alternative = "lower" and 1000 random permutations) are reported. Principal components 1 (PC1) and 2 (PC2) were flipped in panel **d** to facilitate the comparison of climatic axes shown in panel **c**. Data underlying all components of Fig. 3 are provided at https://doi.org/10.5281/zenodo.14988067. Background maps were obtained from the rnaturalearth v.0.3.2 R package.

(2) the outgroup branch rooting the network within the Eastern lineage (Supplementary Fig. 10) points to an Asian origin for the species. We also inferred substantial contemporary gene flow among populations in the Western lineage, as well as some gene flow from the Eastern to the Western lineage (Fig. 2c), the latter consistent with low levels of genetic differentiation between the two lineages ($\Phi_{ST}$ = 0.03 – 0.05; Supplementary Fig. 4). Hence, we uncovered two genetically differentiated lineages and further fine-scale population substructuring, despite substantial gene flow among populations within each lineage.

**Ecological differentiation between Western and Eastern lineages**
We next assessed whether the two genetic lineages are ecologically differentiated. Firstly, the two lineages showed very little overlap in their predicted breeding and non-breeding distributions obtained through species distribution models (SDMs) based on selected bioclimatic variables, reflecting clear differences in lineage-specific

climate relationships (Fig. 3a; Supplementary Fig. 12). Secondly, we detected marked differences in both habitat use and selection between lineages during the breeding season (Fig. 3b). Lesser kestrels primarily rely on herbaceous vegetation, but birds from the Western lineage predominantly used and preferred croplands during breeding, while birds from the Eastern lineage avoided this habitat. The two lineages also differed in their use of urban areas. While birds from the Western lineage used urban areas for nesting[32] (in proportion to their availability), birds from the Eastern lineage used them to a much lesser extent, resulting in avoidance of these habitats. Birds from the Western lineage therefore showed a stronger association with human-modified environments. Thirdly, climatic niche differentiation analyses showed a clear divergence in selected climatic conditions between the two lineages in both breeding and non-breeding ranges, with almost no climatic niche overlap (breeding distribution: Schoener's $D$ = 0.0008; non-breeding distribution: Schoener's $D$ = 0.0012; both equivalency tests $p < 0.001$; Fig. 3c, d). The Western lineage settled in warmer and

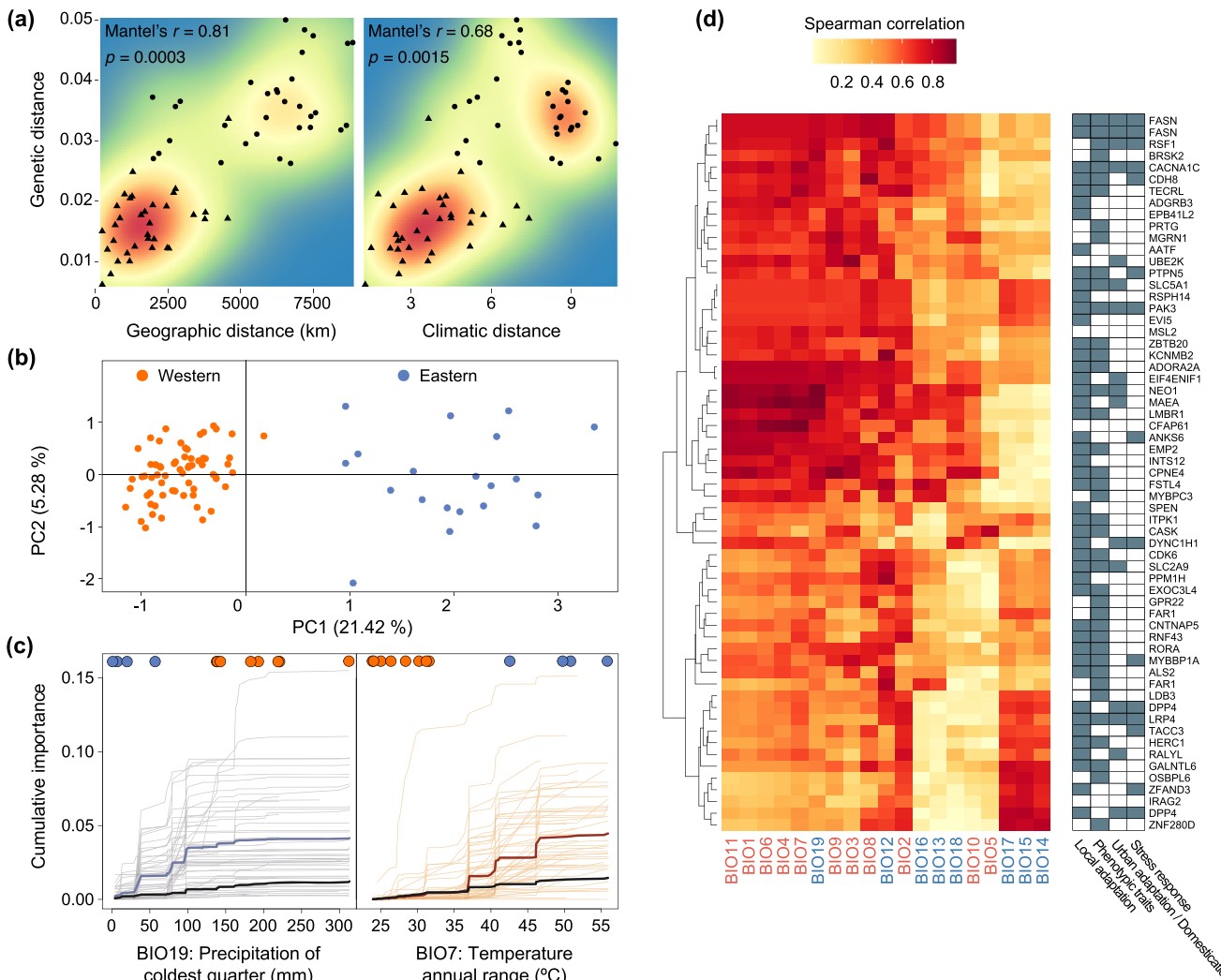

**Fig. 4 | Geography and climate explain spatial genomic variation of lesser kestrels. a** Pairwise genetic distance ($\Phi_{ST}/1 - \Phi_{ST}$) correlates positively with geographic and climatic distance. Mantel tests and their associated *p*-values (one-sided) are reported. Background colours reflect the density of points (blue indicating low density and red indicating high density) and show a discontinuity consistent with a scenario of two distant and differentiated genetic clusters. Distances between localities in the same or different evolutionarily significant units (ESUs) are shown as triangles and circles, respectively. **b** Principal component analysis (PCA) of 61 climate-associated single-nucleotide polymorphisms (SNPs) showing that the two putative adaptive units (AUs) coincide with the identified evolutionarily significant units (ESUs). **c** Allelic turnover functions relative to the two highest ranking bioclimatic variables from a gradient forest (GF) analysis of 61 climate-associated SNPs, *i.e.*, precipitation of the coldest quarter (BIO19) and temperature annual range (BIO7). Y-axis values report the cumulative importance of SNPs in the GF models, which reflects the total amount of allele frequency turnover across the environmental gradient. Thin lines show allelic turnover functions for each of the 61

candidate SNPs. Thick blue and red lines show allelic turnover functions across all candidate SNPs and thick black lines across all putatively non-adaptive reference SNPs. Higher turnover values for candidate SNPs compared to neutral SNPs evidence the stronger association of candidate SNPs with climate. Circles at the top represent sampling localities coloured based on the ESU they belong to (orange: Western; blue: Eastern) ordered along the BIO19 and BIO7 gradients. **d** (Left) Hierarchical clustering of associations between bioclimatic variables (columns) and allele frequencies for the 61 climate-associated SNPs (rows) (Spearman correlation, absolute values). Bioclimatic variables associated with temperature and precipitation are coloured in red and blue, respectively. (Right) Tiles are coloured in grey when the candidate SNP is found within a gene identified, through a literature search (Methods), as being related to local adaptation (first column), phenotypic traits important for local adaptation (second column), adaptation to urban environments or domestication (third column), and/or stress response (fourth column) in vertebrates. Data underlying all components of Fig. 4 are provided at https://doi.org/10.5281/zenodo.14988067.

lower-elevation areas across seasons and in more humid areas during the breeding season compared to the Eastern one (Fig. 3c, d; Supplementary Fig. 13). Overall, these results denote significant ecological differentiation between the two genetically differentiated lineages, which can therefore be regarded as distinct evolutionarily significant units (ESUs; *i.e.*, evolutionarily and ecologically distinct [groups of] populations[33,34]).

### Assessing genetic adaptation to local environments

The two ESUs were also differentiated based on climate-associated genetic variation. Genetic distances strongly correlated with

geographic distances among sampling localities ($r_M = 0.81$, $p < 0.001$; Fig. 4a). However, the correlation was largely driven by the differentiation between the Western and Eastern ESUs, as indicated by non-significant IBD within each ESU ($r_M < 0.49$, $p > 0.06$). Genetic distances were also strongly correlated with climatic distances ($r_M = 0.68$, $p = 0.001$; Fig. 4a), consistent with isolation-by-climate (IBC). When modelling IBD and IBC together using multiple regression on distance matrices (MRM), IBC was no longer statistically significant ($p = 0.56$) while IBD remained significant ($p < 0.001$), although this could be confounded by the strong covariance between geographic and climatic distances ($r_M = 0.79$, $p < 0.001$).

Using a redundancy analysis (RDA) to detect climate-associated SNPs, we uncovered 107 SNPs (after removing those that were more associated with either population structure or geography than with climate). Employing scans of SNPs showing high differentiation between the two ESUs, we identified an additional 11 outlier SNPs potentially under selection. Combining these two sets of SNPs, 61 were located within 58 protein-coding genes and constituted the final list of climate-associated SNPs, which were evenly distributed across the genome (Supplementary Fig. 15). We focused our analyses on SNPs within protein-coding genes because these were more likely to be in linkage with genetic variation potentially associated with climate adaptation. By applying Principal Component Analysis (PCA) to the climate-associated SNPs, we identified a main discontinuity in climate-associated genetic variation, which may indicate two adaptive units (AUs)[35], *i.e.*, groups of populations that share similar adaptive traits and environmental adaptations[36] (Fig. 4b). These AUs coincided with the Western and Eastern ESUs, suggesting that the two ESUs are adapted to different local climatic conditions. A gradient forest (GF) analysis applied to the 61 climate-associated SNPs showed that they were more strongly associated with climatic variables than neutral SNPs, as highlighted by the higher turnover values based on the top-ranked bioclimatic variables (BIO19 and BIO7; Supplementary Fig. 16, Fig. 4c; thick blue and red lines compared to thick black lines). The highest steps in the turnover functions occurred between values of the climatic variables that separated populations from the Western and the Eastern ESUs, with allele frequencies in climate-associated SNPs differing substantially between Western and Eastern ESUs (Fig. 4c). In comparison, allele turnover within each ESU was low and showed very similar gradients at climate-associated and neutral SNPs.

Most of the climate-associated SNPs were located in genes that have been previously identified as candidates for local adaptation in vertebrates (Supplementary Data 2) and are associated with processes such as thermoregulation, lipid metabolism, or differences in migratory behaviour (Supplementary Table 4), potentially contributing to the divergence between Western and Eastern ESUs[37–39]. Hierarchical clustering of absolute Spearman's correlation coefficients between the allele frequencies of the 61 climate-associated SNPs and the values for 19 bioclimatic variables across the sampled populations[40] highlighted clusters of SNPs that covaried in the strengths of correlations with bioclimatic variables (Fig. 4d). Hence, climate-associated SNPs were associated with different environmental gradients, suggesting they might be involved in adaptation to different environments. Taken together, these results highlight that the two genetically and ecologically distinct lineages show evidence of adaptation to their specific environments.

## Effects of past climatic fluctuations on demographic history and changes in distribution range

To provide historical context to the demographic and distributional effects of climatic fluctuations, we reconstructed the lesser kestrel demographic history using genetic data and hindcasted breeding and non-breeding distribution ranges back to the last interglacial period (140–120 kya). Demographic modelling using DIYABC supported a divergence scenario consistent with phylogenetic analyses (Fig. 2d), including admixture events among populations in the Western ESU (Fig. 5b), with a posterior probability of 0.75. The split time between Western and Eastern ESUs was estimated at 40.4 kya (95% confidence interval [CI]: 15.8-68.7 kya), during the second half of the Last Glacial Period (115–11.7 kya), coinciding with a steady demographic decline inferred by the demographic analysis (Fig. 5c) and a decrease in the extent of breeding range (Fig. 5e, g). The divergence time between the mitochondrial haplogroups A (most frequent in the Western ESU) and B (most frequent in the Eastern ESU) was inferred at 49 kya (41.7–56.3 kya) (Supplementary Fig. 17), corroborating the evidence that the split between the two ESUs took place during the Last Glacial

Period. The increase in temperatures after the Last Glacial Period (Fig. 5a) coincided with an expansion of the breeding range for the Eastern ESU, but not for the Western one (Fig. 5e). Yet concomitant increases in effective population size ($N_e$) were not detected until the end of the Younger Dryas (12.9–11.7 kya) (Fig. 5d). This increase differed in pace and magnitude of change between the two ESUs, being particularly pronounced for the Western ESU and indicative of a demographic expansion, which coincided with: (1) a sharp increase in the extent of the non-breeding range in West Africa (Fig. 5f, g); and (2) admixture events among breeding populations in this ESU (Fig. 5b). Despite the sharp temporal changes in the extent of breeding and non-breeding ranges, no marked latitudinal change emerged (Supplementary Fig. 18). These results suggest a strong effect of past climatic fluctuations on the demographic history of the species and identify climate-driven shifts in the species' historical distribution range, highlighting the species' sensitivity to changing climatic conditions.

## Distributional and genetic responses to future climate

Building on the inferred species' sensitivity to past climatic changes, we subsequently evaluated its potential distributional and genetic responses to future climate scenarios. SDMs forecasted that geographic isolation during the breeding season will increase between the Western and Eastern ESUs in the next decades (up to 2100) (Fig. 6a). In particular, the westernmost part of the Eastern ESU distribution range is predicted to become climatically unsuitable, implying that populations in that region might be particularly vulnerable to contemporary climate warming. Under 'extreme warming' future climate, the breeding and non-breeding ranges are predicted to expand substantially for the warmer-adapted Western ESU and are predicted to shrink for the colder-adapted Eastern ESU (Fig. 6a, b), consistent with their adaptations to different climates. Indeed, these predicted changes are consistent with hindcasted trends in range size through the Holocene, although they show a greater magnitude of change compared to the latter (Fig. 6b). In both ESUs, the climate associated with the current breeding range is expected to markedly shift northwards, and for the Eastern ESU the climate associated with its current non-breeding range is predicted to move considerably southward (Supplementary Fig. 18). Lesser kestrels from the Eastern ESU undertake annual migrations that are approximately three times longer than those from the Western ESU (Fig. 6b), and our models suggest that migration distances for both ESUs have been largely invariant throughout the Holocene. However, migration distances are expected to increase in the near future, especially for Asian populations due to forecasted latitudinal range shifts (Fig. 6b, Supplementary Fig. 18). Yet, alternative non-breeding areas might become available for the Eastern ESU closer to their breeding grounds (e.g. south-east Asia; Supplementary Fig. 19). Similar but less marked range and migration distance changes are expected under 'moderate warming' future climate (Fig. 6b; Supplementary Fig. 18).

Based on the genotype-climate associations across the breeding distribution range, we investigated vulnerability to future climate change using genetic offsets, a measure of how much genetic change is needed by populations to adjust to new climate conditions. Populations exhibiting the greatest genetic offsets are those requiring the highest genetic changes, potentially increasing their climate change vulnerability[41]. Genetic offsets estimated for 2041–2070 based on an 'extreme warming' future climate showed slightly higher values for the Eastern ESU (Fig. 7a). The prediction for 2071–2100 showed an even more marked increase in genetic offset in the Eastern ESU, but nearly no increase in the Western ESU (Fig. 7b). The north-western sectors of the Eastern ESU distribution range showed the highest genetic offsets (Fig. 7c, d), suggesting higher risk of climate maladaptation in these regions. Similar trends, albeit less marked, were observed under 'moderate warming' future climate (Supplementary Fig. 20).

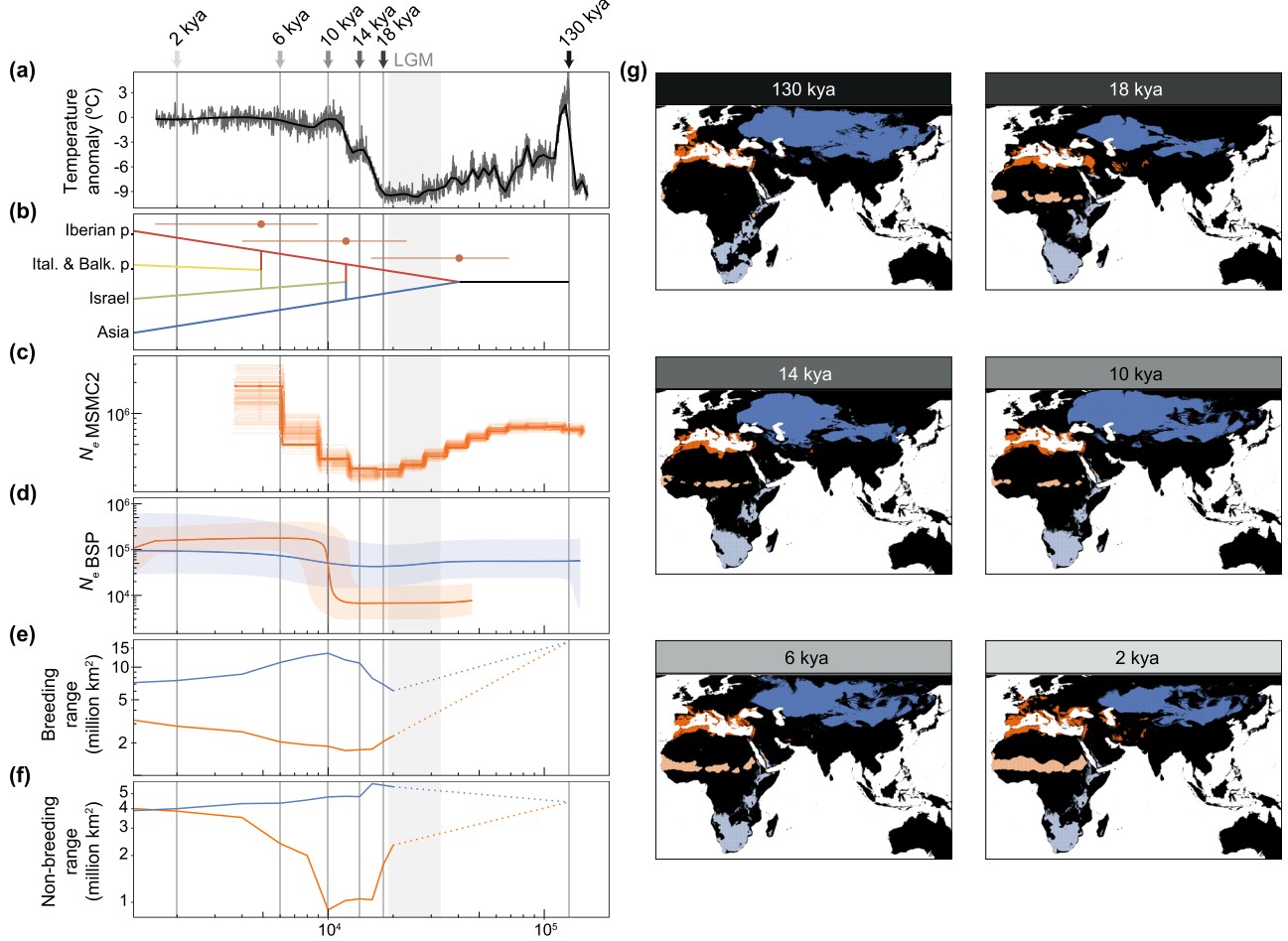

**Fig. 5 | Climate fluctuations, demographic history, and hindcasted breeding and non-breeding distribution ranges of lesser kestrels. a** Temperature anomaly as inferred from the EPICA (European Project for Ice Coring in Antarctica) Dome C ice core[107]; **b** DIYABC best-supported scenario showing divergence and admixture times (mean and 95% confidence interval [CI] from 1000 out-of-bag testing samples using a set of broad priors drawn from uniform distributions [Supplementary Table 9]) of four fine-scale genetic clusters (Iberian peninsula, Italian and Balkan peninsulas, Israel and Asia; Fig. 2b); **c** effective population size ($N_e$) changes through time for the Western evolutionarily significant unit (ESU) estimated by MSMC2 (steps with bootstraps shown as faded steps); **d** Bayesian Skyline Plots (BSP) showing $N_e$ changes through time (ribbons indicating 95% highest posterior density [HPD] intervals with lines indicating the median) obtained from

mitogenome sequences; **e** extent of predicted breeding and **f** non-breeding ranges. The grey shaded area across panels represents the Last Glacial Maximum (LGM). In panels **d**–**f** estimates for the Western and Eastern ESUs are shown in orange and blue, respectively. In panels **e**–**f** estimates are shown for the last 20,000 years (2,000-years steps) and dotted lines connect estimates for 130 kya (when the two ESUs had not diverged yet) to estimates for 20 kya; **g** Predicted breeding and non-breeding ranges for Western and Eastern ESUs at selected timepoints; the timepoints shown in panel **g** are highlighted with grey vertical lines in panels (**a**–**f**). Data underlying all components of Fig. 5 are provided at https://doi.org/10.5281/zenodo.14988067. Background maps were obtained from the rnaturalearth v.0.3.2 R package.

Altogether, these results reveal divergent responses to contemporary climate change between ESUs.

## Discussion

Anticipating species' responses to climate change in an altered and rapidly changing biosphere is pivotal for setting conservation priorities and planning proactive and effective conservation actions. We demonstrate how integrating spatiotemporal ecological and evolutionary dynamics across the annual distribution range of a long-distance migratory species may help in predicting range-wide responses to climate change.

Our study uncovered significant ecological differentiation between two genetically differentiated lineages that we delineate as ESUs, corroborating previous studies[31,42,43]. Populations from the Eastern ESU bred in colder, drier, and higher-elevation areas and showed a weaker association with human-modified environments (croplands, urban areas) than those from the Western ESU.

Accordingly, we identified a main break in climate-associated genetic variation between the two ESUs, suggesting that they are adapted to different climatic conditions. Indeed, the climate-associated SNPs we identified, notwithstanding covariation between population structure and environmental variation, were located within genes potentially involved in divergent selection between Western and Eastern ESUs. Whilst these genes represent promising candidates, we appreciate that we did not survey the full range of genetic variation associated with climate adaptation, due to, for example, limitations in the ddRAD approach [[44]; but see refs. [45],[46]] and neglecting structural variation[47].

Although many studies have addressed responses to contemporary climate change[4,48,49], there is a dearth of historical context regarding whether similar responses have occurred during past periods of climate change[50,51]. Consistent with the lesser kestrel's overall preference for relatively warm and dry habitats, we inferred decreasing trends in $N_e$ under climate cooling during the Last Glacial Period. Climate cooling also resulted in breeding range contraction

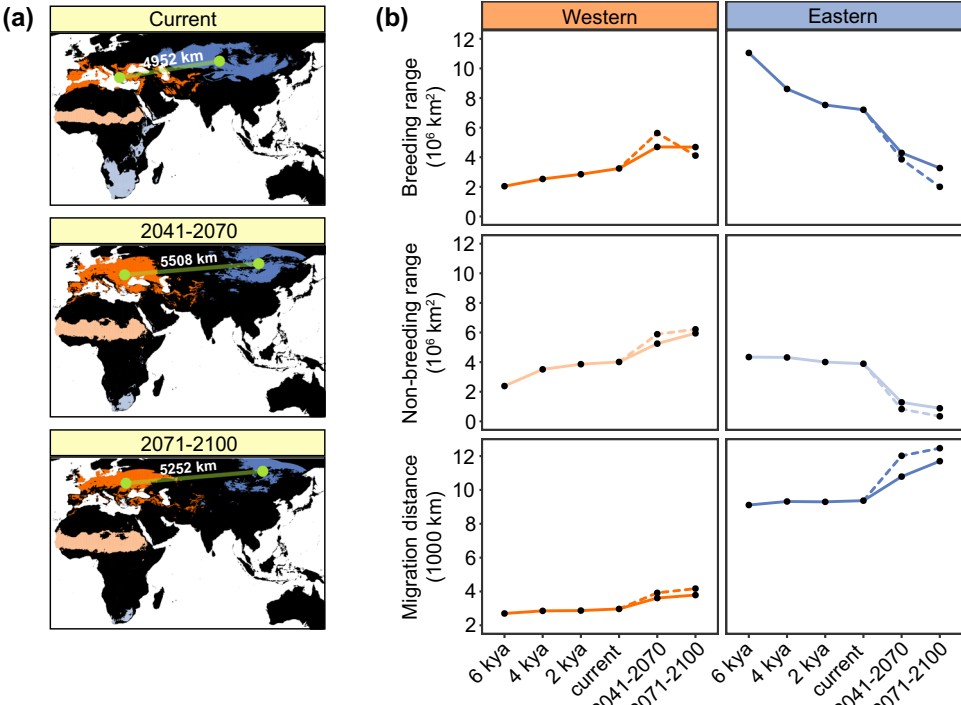

**Fig. 6 | Forecasting lesser kestrel distributional responses to climate change.**
**a** Predicted breeding (dark-shaded colours) and non-breeding (light-shaded colours) range in the present and in the future (2041-2070 and 2071-2100), for Western (orange) and Eastern (blue) evolutionarily significant units (ESUs), using an 'extreme warming' future climate (UKESM1-0-LL; SSP5-8.5). Green circles show the centroids of breeding distributions for each of the ESUs and the distance between centroids is shown above the line connecting them. Geographic isolation between the two ESUs is expected to increase in the future. **b** Trends from 6 kya to the future for: (top) the extent of breeding range, (middle) the extent of non-breeding range,

and (bottom) migratory distance, showing divergent patterns between Western and Eastern ESUs. Migration distance is calculated as the minimum distance between the breeding distribution and the non-breeding distribution range centroids. For future time periods, estimates obtained from the 'extreme warming' future climate are shown as dashed lines, while those from the 'moderate warming' future climate (GFDL-ESM4; SSP3-7.0) are shown as solid lines. Data underlying all components of Fig. 6 are provided at https://doi.org/10.5281/zenodo.14988067. Background maps were obtained from the rnaturalearth v.0.3.2 R package.

and retreats to spatially fragmented refugia in southern portions of the range, such as the Mediterranean region for western populations[52–54]. This range contraction potentially reduced gene flow, leading to genetic differentiation between the Western and Eastern ESUs. We also inferred increasing $N_e$ trends under subsequent climate warming, similar to other warm-adapted species[8]. This increase in $N_e$ was more pronounced for the warmer-adapted Western ESU compared to the colder-adapted Eastern ESU, and coincided with two key events that could have contributed to demographic growth in the Western ESU: (1) a sharp increase in the extent of the non-breeding range, potentially due to major environmental changes in West Africa during the African Humid Period (14,500-5,500 ya)[55], and (2) the onset and rapid spread of agricultural practices in the Middle East and the Mediterranean Basin[56]. Given that a migratory lifestyle inherently provides an ability to track changes in habitat availability, and that novel migratory strategies can emerge within relatively short timeframes[15,16], it appears that lesser kestrels from the Western ESU switched from a longer migratory route to South Africa to a novel and shorter migratory route to West Africa at the beginning of the Holocene. Shorter migration distances could have increased fitness[11,57] and contributed to the inferred increase in $N_e$[58]. Our results highlight the potential for migration to track changing environments, to mediate demographic responses and to promote intraspecific differentiation[14,58].

In Western Europe and the Middle East, lesser kestrels largely rely on agricultural landscapes for foraging[59,60] and on human settlements, including urban areas, for nesting[32], while in Asia, they rely on natural cliffs, sparse rocks on grasslands, and abandoned human settlements[61]. In the Mediterranean region, lesser kestrels have been breeding in

human-modified environments for millennia[32]. Based on our results, we hypothesise that their association with humans likely began with the onset of agricultural practices in this region, which provided novel foraging and nesting opportunities[62]. The exploitation of these new habitats could have triggered the demographic expansion of the Western ESU.

Disregarding intraspecific climatic niche variation can lead to inaccurate predictions of species' vulnerability to climate change[9,10,63]. Our findings, showing divergent climate change responses in two ESUs, align with this idea. The Eastern ESU has experienced a reduction in both its breeding and non-breeding ranges since the onset of the Holocene, and these ranges may shrink at a faster pace in the near future. This ESU, and especially the warm and low elevation populations neighbouring the contact zone with the Western ESU, also showed the highest genetic offsets, suggesting that they could be at risk of climate maladaptation. However, this potential climate maladaptation seems primarily driven by expected climate warming in these areas. Since the Western ESU is currently adapted to warmer environments than the Eastern ESU, the introduction of adaptive variation via gene flow from the former to the latter might make populations more resistant to warmer future climates, avoiding local extinctions[64,65]. Furthermore, the long migration distance of the Eastern ESU is expected to increase in the foreseeable future, unless a switch to novel non-breeding areas, located closer to the breeding sites, occurs. Lesser kestrels are able to track changes in habitat availability and establish new migratory routes, as witnessed by the recent increase of individuals overwintering in southern Europe, although still in small numbers[29,66,67]. Limited monitoring efforts in its breeding grounds and a lack of long-term trend data[30] make it difficult

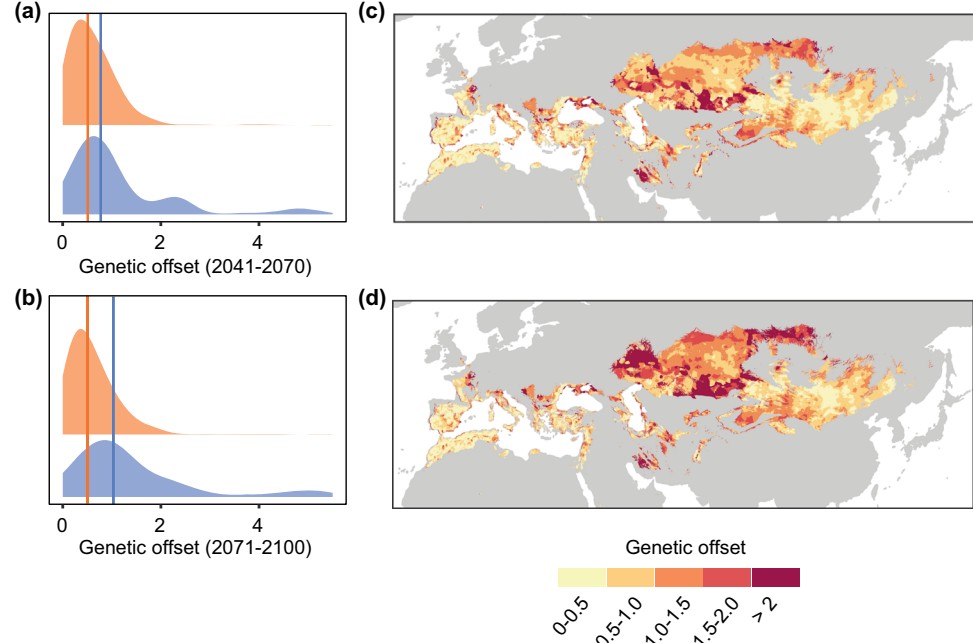

**Fig. 7 | Genetic offsets for lesser kestrel evolutionary lineages. a**, **b** Density plots of genetic offsets in 2041–2070 (**a**) and 2071–2100 (**b**) for all 2.5 arc-minute grid cells with breeding occurrence records within the Western (orange) and Eastern (blue) evolutionarily significant units (ESUs), showing higher offsets for the Eastern ESU. Median values for each ESU are shown as coloured vertical lines. **c**–**d** Genetic offsets across the current breeding range in 2041–2070 (**c**) and 2071–2100 (**d**) based on projections using an 'extreme warming' future climate (UKESM1-0-LL; SSP5-8.5; estimates for a 'moderate warming' future climate are shown in Supplementary Fig. 20). The highest offsets are in the western and northern sectors of the Eastern ESU's current range. Data underlying all components of Fig. 7 are provided at https://doi.org/10.5281/zenodo.14988067. Background maps were obtained from the rnaturalearth v.0.3.2 R package.

to assess if these threats are already resulting in population declines in the Eastern ESU. Yet, non-breeding numbers of lesser kestrels in South Africa have been markedly decreasing in recent years, suggesting population decline and range contraction[68]. The contemporary gene flow out of Asia towards Europe could also be a response to worsening climatic conditions in Asia, particularly in the western sector of the Eastern ESU's breeding range.

Our results also suggest that the Western ESU will expand to higher latitudes in the near future, a pattern supported by the recent colonisation of northern Italy[29]. In this scenario, gene flow from the Eastern ESU, which already breeds at higher latitudes, could introduce adaptive alleles[69,70]. Although the Western ESU might not be immediately threatened by climate change, a strong reliance on agroecosystems for foraging[59,60,71] makes it highly vulnerable to changes in land use and farming practices[72]. Moreover, interannual survival and population size of the Western ESU, similarly to most trans-Saharan migrants[73], depends on interannual variation in Sahel rainfall[74,75]. Indeed, prolonged extreme drought in the Sahel region likely contributed to a dramatic population collapse (above 90%) in the second half of the 20th century[28]. More recently, the southernmost populations of this ESU, which currently represent a species' stronghold[28], are facing an increase in the frequency of heatwaves, which can induce breeding failure, reducing local recruitment and threatening their local persistence[71,76,77]. Since these populations live at the edge of the species' climate tolerance, in areas where climatic conditions are becoming more extreme, they may lack the reservoir of genetic variants that would allow for rapid adaptation to a more extreme climate. However, considering the apparent range flexibility inferred under past climate change, populations might relocate further north, where climatic conditions are expected to become progressively more suitable in the near future. Phenological adjustments (*i.e.*, advances) in the timing of reproduction to avoid breeding with extreme climatic conditions (which may heavily decrease fitness, e.g. late spring/early summer heatwaves causing extensive nestling mortality[71,78]), along with the

potential for phenotypic plasticity[79], represent promising avenues for adaptation to future climate. However, despite the recent partial population recovery of the Western ESU fostered by targeted conservation actions[80], synergies between climate change and habitat degradation in both breeding and non-breeding ranges can pose unforeseen challenges[81].

Our study provides a comprehensive assessment of range-wide spatiotemporal evolutionary and ecological responses to climatic fluctuations in a long-distance migratory bird. The migratory lifestyle allows effective exploitation of temporal shifts in habitat and resource availability, possibly contributing to demographic expansion under favourable climatic conditions by facilitating fast colonisation of new suitable areas. Yet our results revealed that contemporary climate change may challenge the future persistence even of highly mobile taxa, threatening genetic lineages and jeopardising species' adaptability to changing ecological conditions. The resilience of highly mobile species to a rapidly changing climate will ultimately depend on their ability to adaptively respond to novel conditions through different mechanisms, including dispersal, phenotypic plasticity and genetic adaptation.

## Methods
### Range-wide sampling and DNA extraction
All sampling conducted for this study complies with local relevant regulations (details of authorisations are reported in the Acknowledgements). We collected blood (approx. 50 µl) from 119 unrelated nestlings and breeding adults at 16 localities across the whole breeding range (Fig. 1a, Supplementary Table 1, Supplementary Data 1). Blood was stored either in 100% EtOH at 4 °C or on blood storage cards (NucleoCards, Macherey-Nagel). We extracted DNA from blood samples using the NucleoSpin Tissue Kit (Macherey-Nagel) or Qiagen DNeasy Blood and Tissue Kit (QIAGEN). DNA quantity and purity were assessed using a Nanodrop 2000 Spectrophotometer or a Qubit 2.0 Fluorometer (Thermo Fisher Scientific).

## Genomic data generation

**Overview of genomic data generation and sample size.** We generated whole-genome sequencing (WGS) data for 7 individuals (including one female offspring and its parents), double-digest Restriction-Site Associated DNA (ddRAD) data for 84 unrelated individuals, and mitogenome sequencing data for 92 individuals (Supplementary Table 1). As detailed in the following sections, WGS data were used to assemble a reference genome, estimate genome-wide heterozygosity, and reconstruct past demographic history (ca. 130-6 kya), while ddRAD and mitogenome sequencing data were used to perform analyses of population structure. ddRAD data were further used to assess AUs and genomic vulnerability, and mitogenome data to reconstruct the recent demographic history.

**Genome assembly, ddRAD sequencing and genotyping.** As part of the Vertebrate Genomes Project (https://vertebrategenomesproject.org/), we generated a reference-quality genome using the trio-binning pipeline[82,83]. Briefly, this approach relies on the generation of short reads from the parental genomes of a diploid species to partition long-reads from the offspring (in this case, a heterogametic female) into haplotype-specific sets, followed by independent assembly of both maternal and paternal haplotypes (Supplementary Methods 1). During curation, the paternal assembly was chosen as the representative one and the W chromosome was added to it from the maternal assembly. The final genome assembly was 1.22 Gb long and fully phased. Scaffold N50 was 91.8 Mb and 98.92% of the genomic sequence was assigned to 25 autosomes, the Z and W chromosomes, and the mitogenome (Supplementary Fig. 1; Supplementary Table 2). The paternal assembly was functionally annotated using the Eukaryotic Genome Annotation Pipeline v.8, and was screened for repetitive elements using a combination of Windowmasker v1.0.1 and RepeatMasker 4.1.0 with Dfam_3.1 (profile HMM library), and Rebase version 20170127. Functional completeness was evaluated using BUSCO 4.1.4. Genome annotation identified 19,775 genes and pseudogenes, including 16,079 protein-coding, as well as 39,936 fully supported coding sequences (see NCBI annotation at https://www.ncbi.nlm.nih.gov/datasets/gene/GCF_017639655.2/).

Details of library preparation and sequencing of ddRAD markers are reported in Supplementary Methods 2. Reads were demultiplexed using process_radtags in Stacks v2.0. Demultiplexed reads were aligned to the reference genome using BWA mem v.0.7.17 with default parameters and processed with SAMtools v.1.10. The average per-sample depth of coverage was 45× (min-max: 16-68×). To obtain high-quality SNPs, variant calling was implemented using two independent methods, retaining only SNPs identified by both (Supplementary Methods 2). The final dataset comprised 73,373 SNPs genotyped across 84 individuals.

**Mitogenome data generation.** Mitogenomes were generated by amplifying three long-range overlapping fragments using PCR and preparing genomic libraries that were sequenced paired-end on an Illumina MiSeq[84] (Supplementary Methods 2). Reads were demultiplexed using the Illumina bcl2fastq2 Conversion Software v2.20 and subsequently cleaned using TrimGalore v0.6.4. We used BWA mem v0.7.17 to align reads to the lesser kestrel reference mitogenome, obtained using the mitoVGP pipeline (Supplementary Fig. 2). Mitogenomes were sequenced at a mean coverage of 719× (± 256 s.d.). Variant calling was performed using Geneious v8.0.5.

**WGS data generation and SNP calling.** Short-read genomic libraries for three individuals from Tuva (Russia) and one individual from Matera (Italy) were prepared using the Illumina PCR-Free library preparation kit. Libraries were sequenced paired-end (2 × 150 bp) on a NovaSeq 6000 with a target coverage of 20×.

WGS data for these four individuals were combined with WGS data for the male and female (from Matera) used for generating the reference genome (Supplementary Methods 1). Hence, for analyses involving WGS data, we used data derived from either three unrelated individuals from the Western ESU (demographic reconstruction with MSMC2) or six unrelated individuals (three from the Western and three from the Eastern ESU; estimation of individual heterozygosity). Reads were cleaned using TrimGalore v.0.6.7. Reads were aligned to the lesser kestrel genome using BWA mem v0.7.17 and processed and sorted with SAMtools v1.10. Duplicate reads were removed using Picard tools v2.18.29 MarkDuplicates. Variants were called with SAMtools mpileup and BCFtools call; the bamCaller.py script (provided in the msmc-tools package; https://github.com/stschiff/msmc-tools) was used to generate sample-specific variant call format (VCF) and callability mask files.

**Population structure, gene flow and ESU identification.** To explore the range-wide patterns of genetic structure, we applied a PCA implemented in PLINK v.1.9 to a linkage-pruned dataset of SNPs from ddRAD data ($n = 27,853$ SNPs), upon checking the robustness of this dataset to detect the primary axis of population differentiation (Supplementary Methods 3). We further calculated individual ancestries according to a maximum-likelihood (ML) model-based clustering analysis using Admixture v.1.3, with $K$ ranging from 1 to 15 and default parameters. Model fit was evaluated by cross-validation. Fine-scale population structure was evaluated using fineRADstructure v.0.3. We ran the model for 100,000 Markov chain Monte Carlo (MCMC) iterations with a burn-in of 100,000 iterations, and sampling every 1000 iterations. A tree was constructed with 10,000 hill-climbing iterations and the results were visualised using the scripts fineRADstructurePlot.R and finestructureLibrary.R (https://github.com/millanek/fineRADstructure).

To investigate patterns of gene flow among sampling localities, we used the Estimated Effective Migration Surfaces (EEMS) approach (Supplementary Methods 3). We also estimated contemporary gene flow among the four main clusters inferred by fineRADStructure (Iberian peninsula, Italian and Balkan peninsulas, Israel and Asia) using BayesAss3-SNPs (Supplementary Methods 3).

ESUs were delineated following established methods[35]. Briefly, we combined results from the PCA, Admixture and EEMS analyses to highlight discontinuities in genetic variation across the breeding range.

To visualise genealogical patterns, we built a maximum-likelihood population tree with Treemix v.1.13, which uses allele frequency co-variances as well as a Gaussian approximation to assess genetic drift in blocks of 500 SNPs. The common kestrel (*Falco tinnunculus*) was used as an outgroup. We simulated ddRAD data for this species using RADinitio and its reference genome[85]. We also used SplitsTree5 v.5.0.16 to infer a Neighbour-net phylogenetic network using a genetic distance matrix.

Pairwise $\Phi_{ST}$[86] among localities with more than three individuals with ddRAD data ($n = 12$ sampling localities; three localities with < 3 individuals were excluded to improve the accuracy of inter-population comparisons) was calculated using the populations module in Stacks. This measure of population differentiation was chosen due to its robustness to low sample sizes[86].

A mitochondrial coding-region (15,663 bp) Maximum Parsimony (MP) tree was reconstructed based on the reference mitogenome using mtPhyl v.5.003, specifically modified for the analysis of lesser kestrel mitogenomes. Indels were not considered for tree construction. Mitogenomes were aligned using the Geneious progressive pairwise alignment algorithm and the tree was rooted using the available common kestrel reference mitogenome (NC_011307). In addition, haplotype genealogy graphs were constructed and visualised in Fitchi v.1.1.4 (−e 1 option).

To characterise range-wide patterns of heterozygosity, we calculated the proportion of heterozygous sites for each individual using the ddRAD dataset with no missing data (36,932 SNPs).

**Modelling current, past and future distribution and assessing ecological differentiation.** We relied on species distribution models (SDMs) to reconstruct the current breeding and non-breeding distribution of the lesser kestrel, and used these models to infer both past and future distribution, assuming consistent lineage-specific climate relationships across time and no effects of interspecific competition. The latter assumption is based on the fact that lesser kestrels largely co-occur with other ecologically and phylogenetically related species (e.g. *F. vespertinus*, *F. tinnunculus*) across their breeding distribution range[87], often syntopically[88]. To model distribution, we relied on a dataset of breeding occurrence records of lesser kestrels in Eurasia and of non-breeding occurrence records in Africa, obtained from different sources (Supplementary Methods 4). SDMs were implemented using MaxEnt, a presence-background algorithm which has been shown to outperform other methods for modelling occurrence-only data[89]. Further details about MaxEnt model parametrisation, choice of background locations, and reliability checks are reported in the Supplementary Methods 4.

As climatic predictors of distributions, we considered four variables retrieved from the CHELSA v. 2.1 database (period 1981–2010, 2.5 arc-minutes spatial resolution), *i.e.* mean annual air temperature (BIO1), annual range of air temperature (BIO7), sum of annual precipitation (BIO12), and precipitation seasonality (BIO15). These variables represent both average conditions and their variability across the year, are major determinants of vertebrate distributions[90], and account for most of the climatic variation at the global scale[91].

SDMs were built separately for each of the two ESUs (Western and Eastern), and for the breeding and non-breeding season. Non-breeding records in southern Africa were attributed to the Eastern ESU while those from central Africa were attributed to the Western ESU, based on evidence from mitochondrial DNA haplotypes, major histocompatibility complex genes, ring recoveries, and individual tracking studies[42,43,92,93] (Fig. 1b). Breeding records from Portugal to Iran were assigned to the Western ESU, whereas those from Kazakhstan eastwards were attributed to the Eastern ESU, according to the population structure results (Fig. 1b). As a proxy of the distribution range, we chose those cells with suitability values higher than specific thresholds that returned the best concordance between predicted and known distributions (Supplementary Fig. 14 and Supplementary Methods 4). Base layers for maps were plotted using the R package rnaturalearth v.0.3.2.

Values for past and future climate were obtained from WorldClim (last interglacial period, approx. 120,000–140,000 years BP), CHELSA v.2.1 (2,000-year steps for the last 20,000 years) and Coupled Model Intercomparison Project Phase 6 (CMIP6, periods 2041–2070 and 2071–2100). For future climate, we used the two top-priority climate models (a 'warmer' and a 'milder' model) recommended by the Intersectoral Impact model Intercomparison Project (ISIMIP) for climate change impact assessments, namely UKESM1-0-LL and GFDL-ESM4. For the 'warmer' model, we present results for a 'worst case' scenario (SSP5-8.5) to represent extreme warming conditions ('extreme warming' future climate). For the 'milder' model, we present results for a 'business-as-usual' scenario (SSP3-7.0) to represent moderate warming conditions ('moderate warming' future climate). All variables were downloaded at a 2.5 arc-minutes resolution. To estimate the potential past/future ranges (for each combination of ESU, season and past/future period), we limited the extent of suitable areas within a 1500 km-buffer around known occurrence locations; this distance was selected to exclude areas highly unlikely to ever be occupied by the species based on current evidence.

We investigated landscape-scale habitat preferences of lesser kestrels from the Western and Eastern ESUs at their breeding and non-breeding ranges using high-resolution land cover data obtained from Copernicus Land Monitoring Service (https://land.copernicus.eu/en). Habitat use was calculated as the proportion of seven main land cover categories (forest, shrubs, herbaceous vegetation, bare ground, cropland, urbanised, other; pooled from 22 original land use categories; Supplementary Table 3) in all 2.5 arc-minute grid cell with lesser kestrel breeding (Eurasia) or non-breeding (Africa) occurrence records. Habitat availability was estimated as the proportion of different land cover categories within the above-mentioned 1,500 km-buffer, separately for each combination of ESU and season. To assess habitat preferences, we relied on a permutation-based approach comparing the proportion of used and available land cover categories using sign tests[94]. For each 2.5 arc-minute grid cell with occurrence records, we also retrieved point elevation (m a.s.l.) using the get_elev_point function from the R elevatr package.

To evaluate climatic niche differentiation between the two ESUs in their breeding and non-breeding ranges, we relied on an ordination-based approach using all 19 bioclimatic variables from the CHELSA v2.1 database (Supplementary Table 5). We first applied a PCA to breeding and non-breeding climatic conditions available for each ESU (climatic conditions estimated at background locations scattered within an area defined by the above-mentioned 1,500 km-buffer around occurrence records). Then, PCA scores for occurrence records of both ESUs, representing climatic niches to be compared, were projected onto a two-dimensional niche space defined by the first two PCs, separately for breeding and non-breeding ranges. We estimated the extent of niche overlap using the Schoener's *D* metric as implemented in the ecospat R package v.4.0.0, which ranges from 0 (no overlap) to 1 (complete overlap)[95]. We tested for niche divergence between the two ESUs at both seasons with equivalency tests, as implemented in the ecospat.niche.equivalency.test function from the ecospat R package (option overlap.alternative = "lower" and 1,000 random permutations).

**Detection of loci associated with climatic variation, AUs identification and genetic offsets**
**Analysing patterns of isolation-by-distance and isolation-by-climate.** We investigated patterns of isolation-by-distance (IBD) and isolation-by-climate (IBC) by testing the associations between pairwise genetic ($\Phi_{ST}/(1 - \Phi_{ST})$), geographic, and climatic distance matrices between sampling localities using Mantel tests (R package vegan v.2.6-4; $n = 12$ localities). Climatic distances were calculated as Euclidean distance across all PCs from a PCA of all scaled bioclimatic variables ($n = 19$ variables, BIO1-BIO19, Supplementary Table 5) downloaded from CHELSA v.2.1 for the 1981–2010 period at a 2.5 arc-minutes resolution for each sampling locality. We also performed a multiple regression on distance matrices (MRM; R package ecodist v.2.1.3) to control for geographic distance when testing for IBC.

**Assessing climate-associated genetic variation and identifying AUs.** To identify climate-associated SNPs, we employed a two-step approach using genotype-environment association analysis (GEA) and methods to detect differentiation outliers. First, we performed a RDA, a multivariate GEA that is a form of constrained ordination. We chose RDA over alternative GEA approaches because it achieves lower false positive rates[96]. Climatic predictors included the 19 bioclimatic variables (Supplementary Table 5) for each sampling locality. Because we were interested in the detection of climate-associated SNPs rather than SNPs associated with a particular climatic predictor, we performed PCA of the bioclimatic variables to avoid the inclusion of highly-correlated predictors, and we retained the first

three principal components, based on inspection of a scree plot. We ran the RDA using the vegan R package with 37,411 SNPs and 12 sampling localities (after removing SNPs with a minimum allele frequency < 0.05 and sampling localities with fewer than three sampled individuals). We tested the significance of the constrained axes and of the overall model using ANOVA with 9,999 permutations. We identified putative climate-associated SNPs as those with loadings of ± 3 s.d. from the mean score ($p = 0.003$) of the first constrained axis (RDA1), which was the only statistically significant axis. Because loadings of sampling localities on RDA1 were very similar to the main axis of population structure and longitudinal variation, we employed a further strategy to filter SNPs that were more associated with either population structure or geography than with climate (Supplementary Methods 5). Second, we performed two differentiation-based analyses using OutFLANK v.0.2 and PCAdapt v.4.3.3. Loci detected as outliers by both OutFLANK and PCAdapt were considered outliers as they are more likely to be under selection than outlier loci identified by a single method[97] (Supplementary Methods 5). Merged RDA and differentiation outliers were annotated using SnpEff and we retained only SNPs located within genes, as they are more likely to be functional (or in linkage with functional variants).

To examine the relative importance of environmental variables in explaining genetic variation and to determine the turnover of allele frequencies across climatic gradients, we applied a GF approach[98,99]. GF uses a machine-learning regression tree to identify climatic gradients associated with genetic variation to build turnover functions that show how allele frequencies change across a given climatic gradient. We used the 61 climate-associated SNPs and 36,122 neutral SNPs as response variables and the 19 bioclimatic variables as predictors (Supplementary Table 5). Important gradients have greater overall cumulative importance (total amount of turnover in allele frequency across the climatic gradient) and steeper changes in the turnover function denote rapid changes in allele frequencies[99].

To further visualise patterns of associations between allele frequencies and climatic gradients in the 61 climate-associated SNPs, we calculated Spearman's correlation coefficients between allele frequencies per sampling locality and each of the 19 bioclimatic variables. The heatmap.2 function from the gplots v.3.1.3 R package was used to visualise the clustering of SNPs based on their correlations with bioclimatic variables.

To explore the potential roles of genes with climate-associated SNPs in local climate adaptation, we conducted literature searches in Web of Science (www.webofscience.com) and Google Scholar (scholar.google.com) with the keywords: (1) "local adaptation", (2) "phenotypic traits", (3) "domestication/urban adaptation" and (4) "stress response". We selected articles that focused on vertebrate species and that reported associations between the candidate genes and the keywords.

AUs were delineated based on discontinuities in climate-associated genetic variation across the breeding range, according to the criteria proposed by Turbek et al.[35]. To identify climate-associated genetic structure, we performed a PCA using only climate-associated SNPs.

**Genetic offsets under future climatic conditions.** Genetic offsets leverage genomic information to forecast climate maladaptation, and represent the magnitude of the required changes in allelic composition that would allow populations to keep pace with predicted changes between present and future climate[41]. The rationale behind genetic offsets is to predict potential shifts in adaptive optimum induced by climate change and identify geographical regions within the distribution range where the predicted allelic frequency shifts might be too large for populations to adapt to the new conditions. Large shifts could result in maladaptation, reducing fitness and potentially resulting in demographic declines[100–102]. We predicted shifts in allele frequencies of climate-associated SNPs induced by climate change using a genetic

offset statistic based on RDA[103]. We followed the approach outlined in Capblancq & Forester[103] to calculate an adaptive index across the lesser kestrel breeding distribution range (current range, estimated with SDMs) for each grid cell (2.5 arc-minutes resolution) using the first RDA axis of an 'adaptively enriched' RDA[104]. Such an RDA was built by including the 61 climate-associated SNPs as the multivariate response and five bioclimatic variables (BIO19, BIO8, BIO2, BIO15, BIO10) as independent predictors. The latter were selected based on their relative importance in the GF analysis, starting from the highest ranking variable and keeping those variables showing limited collinearity with previously selected ones (i.e., $|r| < 0.7$). The adaptive index was also calculated for 2040–2071 and 2071–2100 under both 'extreme warming' and 'moderate warming' future climates. We calculated the Euclidean distance of the index between current and future climates as a proxy for genetic offset, higher values implying higher risk of maladaptation.

## Demographic history reconstruction

To provide historical context on how past climate change may have affected demographic trends, we used genomic data to reconstruct demographic history. Using ddRAD data, we tested support for different divergence scenarios using DIYABC Random Forest v.1.0. We evaluated four demographic models based on the population structure results using four demes (Iberian peninsula, Italian and Balkan peninsulas, Middle and Eastern Europe, Israel and Asia). These involved (1) a simultaneous split between all demes, (2) a simultaneous split between Asia, Israel and Europe, and further split between Iberia and rest of Europe, (3) a model based on the Treemix topology, (4) a model based on the Treemix topology with an east-to-west expansion. After best-model selection, we tested several admixture scenarios. Population parameters were estimated for the best model (Supplementary Methods 6).

We also used mitogenomes to estimate divergence times among haplogroups using Beast v.2.6.3 assuming a HKY substitution model (gamma-distributed rates plus invariant sites) and a relaxed clock (log-normal). To find the best priors for Bayesian analysis, ML estimations were performed using BaseML in PAMLX v.1.3.1.

To estimate changes in $N_e$ over time, we reconstructed demographic histories using two different approaches: one using WGS data and one using mitogenomes. For mitogenomes, we produced Bayesian skyline plots (BSP; in Tracer v1.7.1) separately for each ESUs (Western and Eastern). The substitution rate was informed by mitogenome phylogenetic analysis in Beast and the mean generation time was set to 2 years[105] (Supplementary Methods 6).

For WGS data, we applied MSMC2 to data from three unrelated individuals from Italy. Haplotypes were inferred using a combination of WhatsHap v1.4 and SHAPEIT4 v4.1.2. The script generate_multihetsep.py from the msmc-tools package was used to merge individual variants and generate an input file for MSMC2. Results were scaled using a mutation rate of $3.3 \times 10^{-9}$/gen/site as estimated for the peregrine falcon (*Falco peregrinus*)[106] and a generation time of 2 years.

## Reporting summary

Further information on research design is available in the Nature Portfolio Reporting Summary linked to this article.

## Data availability

The lesser kestrel reference genome was deposited on the National Centre for Biotechnology Information (NCBI) with accession numbers: GCF_017639655.2 [https://www.ncbi.nlm.nih.gov/datasets/genome/GCF_017639655.2/] (primary) and GCA_017639645.1 [https://www.ncbi.nlm.nih.gov/datasets/genome/GCA_017639645.1/] (alternate). The genome annotation is available on NCBI (NCBI *Falco naumanni* Annotation Release 100 at https://www.ncbi.nlm.nih.gov/refseq/annotation_euk/Falco_naumanni/100/). Population-level ddRAD, mitogenome and WGS data are archived on the European Nucleotide Archive (ENA) (Project

accession number PRJEB71106). Details of genetic samples, breeding and non-breeding occurrence records used for SDMs, and results of literature searches for genes with SNPs associated with climate, are provided as Supplementary Data 1-3. The data underlying the figures are provided at https://github.com/jferrerobiol/lk_climate (https://doi.org/10.5281/zenodo.14988067).

## Code availability

All code, scripts and additional data to reproduce the analyses conducted in this study are available on GitHub: https://github.com/jferrerobiol/lk_climate (https://doi.org/10.5281/zenodo.14988067).

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

## Acknowledgements

We are grateful to the many colleagues, collaborators and fieldworkers who made this study possible, and to J. Johnson (The Peregrine Fund) for constructive criticism that improved a previous draft of the manuscript. M. Pavia kindly provided advice on the past distribution of lesser kestrels in Europe. Most data regarding the distribution of lesser kestrel breeding colonies in Kazakhstan were collected by the late E. Bragin. Permits required for conducting this study were provided by the following local or national authorities (permit references in brackets): Croatian Ministry of Environment (KL: UP/I-612-07/15-48/ 108; URBROJ: 517-07-1-1-1-15-3; from 6 July 2015), Hellenic Ministry of Environment and Energy (ΒΕΝΔ0-ΔΔ8, ΒΛ9Σ0-Γ3Α, ΩΗΛΔ465ΓΘΗ-3ΓΙ), Israel Nature and Parks Authority (2015/40829), ISPRA (Law 157/ 1992 [Art.4 (1) and Art. 7 (5)]), Regione Sicilia (1616/2014), Ministry of Environment and Tourism of Mongolia (06/2564), Institute for Nature Conservation and Forests (ICNF) of Portugal (158/2013, 82/2020), Consejería de Medio Ambiente, Junta de Andalucía, Spain (SGYB-AFR-CMM), General Directorate of Nature Conservation and National Parks (MoAF) of Turkey (21264211-288.04-E.2059415). This study was carried out with partial financial support from the European Union LIFE funding scheme (projects LIFE FALKON, LIFE17 NAT/IT/000586 and LIFE for Lesser Kestrel, LIFE19 NAT/BG/001017) and the PRIN funding scheme of the Italian Ministry for University and Research (grants 20178T2PSW to D.R., 2017CWHLHY to A.T. and L.G., 2022CWMRNH to M.M., the latter funded by the European Union - NextGenerationEU). A.G., A.I., A.O., D.R. and M.M. were supported by the National Recovery and Resilience Plan (NRRP), M4C2 Investment 1.4 - 'Strengthening of research facilities and creation of 'national R&D champions' on some Key Enabling Technologies', funded by the European Union - NextGenerationEU (Project code CN_00000033, National Biodiversity Future Centre - NBFC) under the following project codes (CUP): A.G. and D.R.: H43C22000530001; A.I. and M.M.: B83C22002930006; A.O.: F13C22000720007. Financial support to CE3C was provided by FCT (UID/00329/2025). J.G. was supported by the project NORTE-01-0246-FEDER-000063, Norte Portugal Regional Operational Programme (NORTE2020), under the Portugal 2020 Partnership Agreement through the European Regional Development Fund (ERDF). Any use of trade, firm, or product names is for descriptive purposes only and does not imply endorsement by the U.S. Government.

## Author contributions

J.F.O., A.B., and D.R. conceived the study. J.F.O., A.B., G.L., A.I., G.F., A.B.-A., G.F.F., A.G., J.B., M.C., B.H., J.M., E.-A.T., C.C., L.G., E.D.J., A.O., K.S., A.T., and D.R. provided laboratory facilities, performed laboratory analyses and generated the genomic datasets, including genome assembly and annotation. J.F.O., A.B., M.B., N.B., A.E.B., I.C., J.G.C., B.D., F.D.P., J.G., T.E.K., M.M., and D.R. assembled and curated the geographic distribution and environmental datasets. J.F.O, A.B., M.B., G.L., and S.S.

performed data analysis. J.R.P., J.R.W., G.F., G.F.F., A.O., E.T., A.T., and D.R. assisted with data analysis and interpretation. J.F.O. wrote the paper in collaboration with A.B., M.B., G.L., S.S., J.R.P. and D.R. A.B., N.B., A.E.B., M.C., I.C., J.G.C., B.D., F.D.P., R.E., K.E.-Y., J.G., G.G., T.E.K., K.M., M.M., L.G.P., A.R., M.S., N.T., M.W. and D.R. collected samples for genetic analyses and contributed to writing the manuscript.

## Competing interests
The authors declare no competing interests.

## Additional information

Joan Ferrer Obiol ⬥[1] ✉, Anastasios Bounas ⬥[2], Mattia Brambilla ⬥[1], Gianluca Lombardo ⬥[3,4], Simona Secomandi ⬥[4,5], Josephine R. Paris[6,7], Alessio Iannucci ⬥[8], James R. Whiting ⬥[9], Giulio Formenti ⬥[10], Andrea Bonisoli-Alquati ⬥[11], Gentile Francesco Ficetola ⬥[1], Andrea Galimberti ⬥[12,13], Jennifer Balacco[10], Nyambayar Batbayar ⬥[14], Alexandr E. Bragin ⬥[15], Manuela Caprioli[1], Inês Catry ⬥[16], Jacopo G. Cecere ⬥[17], Batmunkh Davaasuren[14], Federico De Pascalis ⬥[17], Ron Efrat ⬥[18], Kiraz Erciyas-Yavuz ⬥[19], João Gameiro ⬥[20,21,22], Gradimir Gradev ⬥[23,24], Bettina Haase[10], Todd E. Katzner ⬥[25], Jacquelyn Mountcastle[10], Kresimir Mikulic[26], Michelangelo Morganti ⬥[12,27], Liviu G. Pârâu ⬥[28], Airam Rodríguez ⬥[29], Maurizio Sarà ⬥[30], Elisavet-Aspasia Toli ⬥[2], Nikos Tsiopelas[31], Claudio Ciofi ⬥[8], Luca Gianfranceschi ⬥[4], Erich D. Jarvis ⬥[5,10], Anna Olivieri ⬥[3,12], Konstantinos Sotiropoulos ⬥[2], Michael Wink ⬥[28], Emiliano Trucchi[7], Antonio Torroni ⬥[3] & Diego Rubolini ⬥[1] ✉

[1]Dipartimento di Scienze e Politiche Ambientali, Università degli Studi di Milano, Milano, Italy. [2]Department of Biological Applications and Technology, University of Ioannina, Ioannina, Greece. [3]Dipartimento di Biologia e Biotecnologie "Lazzaro Spallanzani", Università degli Studi di Pavia, Pavia, Italy. [4]Dipartimento di Bioscienze, Università degli Studi di Milano, Milan, Italy. [5]Laboratory of Neurogenetics of Language, The Rockefeller University, New York, NY, USA. [6]Dipartimento di Medicina clinica, Sanità pubblica, Scienze della Vita e dell'Ambiente, Università degli Studi dell'Aquila, Coppito, Italy. [7]Dipartimento di Scienze della Vita e dell'Ambiente, Università Politecnica delle Marche, Ancona, Italy. [8]Dipartimento di Biologia, Università degli Studi di Firenze, Sesto Fiorentino, Italy. [9]Department of Biological Sciences, University of Calgary, Calgary, Alberta, Canada. [10]The Vertebrate Genome Laboratory, The Rockefeller University, New York, NY, USA. [11]Department of Biological Sciences, California State Polytechnic University - Pomona, Pomona, CA, USA. [12]National Biodiversity Future Centre (NBFC), Palermo, Italy. [13]Dipartimento di Biotecnologie e Bioscienze, Università degli Studi di Milano-Bicocca, Milan, Italy. [14]Wildlife Science and Conservation Center of Mongolia, Ulaanbaatar, Mongolia. [15]NGO Naurzum, Kostanay, Kazakhstan. [16]Centre for Ecology, Evolution and Environmental Changes (CE3C) & CHANGE – Global Change and Sustainability Institute, Faculdade de Ciências da Universidade de Lisboa, Lisboa, Portugal. [17]Area Avifauna Migratrice, Istituto Superiore per la Protezione e la Ricerca Ambientale, Ozzano dell'Emilia, Italy. [18]Department of Evolutionary and Environmental Biology, University of Haifa, Haifa, Israel. [19]Ornithological Research Center, Ondokuz Mayis University, Samsun, Turkey. [20]CIBIO, Centro de Investigação em Biodiversidade e Recursos Genéticos, InBIO Laboratorio Associado, Campus de Vairão, Universidade do Porto, Vairão, Portugal. [21]CIBIO, Centro de Investigação em Biodiversidade e Recursos Genéticos, InBIO Laboratorio Associado, Instituto Superior de Agronomia, Universidade de Lisboa, Lisbon, Portugal. [22]BIOPOLIS Program in Genomics, Biodiversity and Land Planning, CIBIO, Campus de Vairão, Vairão, Portugal. [23]Green Balkans - Stara Zagora NGO, Stara Zagora, Bulgaria. [24]Department of Agroecology, Agricultural University - Plovdiv, Plovdiv, Bulgaria. [25]U. S. Geological Survey, Boise, ID, USA. [26]IBIS program LTD, Zagreb, Croatia. [27]Consiglio Nazionale delle Ricerche - Istituto di Ricerca Sulle Acque (CNR-IRSA), Brugherio, Italy. [28]Institute of Pharmacy and Molecular Biotechnology, Heidelberg University, Heidelberg, Germany. [29]Departamento de Ecología Evolutiva, Museo Nacional de Ciencias Naturales (MNCN), CSIC, Madrid, Spain. [30]Dipartimento STEBICEF, Università degli Studi di Palermo, Palermo, Italy. [31]Hellenic Ornithological Society, Athens, Greece. ✉e-mail: joan.ferrer.obiol@gmail.com; diego.rubolini@unimi.it

