## [Transparent Peer Review file · Nature Communications]

Evolutionarily distinct lineages of a migratory bird of prey show divergent responses to climate change

Corresponding Author: Dr Joan Ferrer Obiol

Version 0:

Reviewer comments:

Reviewer #1

(Remarks to the Author)

Major comments:

This is a very interesting study on predicting lesser kestrels' responses to past climate fluctuations and future climate changes using landscape genomic and demographic approaches and species distribution models based on de novo genome sequencing, whole-genome sequencing, ddRAD sequencing, and climatic data. The authors showed that the genetic divergences and ecological differences of the European and Asian lineages of the lesser kestrel are apparently attributed to past climatic fluctuations, and the cold-adapted Asian lineage may be at higher extinction risk than the warm-adapted European lineage when they face future dramatic climate changes.

Although the authors did lots of analytical works to test their hypothesis and the entire pipeline is quite complete, I am still concerned whether the ddRAD data with limited SNP number (only 73,373 SNPs) and high missing rate (25% individual without genotypes) can provide enough power to interpret genetic differentiation and climatic adaptation. Perhaps, it is enough for genetic differentiation but not for the scans of adaptive SNPs. Additionally, I am not sure whether a large sampling gap between KAZ and RUS populations would affect the population structure and genetic differentiation analyses. I recommend the authors to supplement samples from a few localities in the gap and do the WGS for all individuals. I don't think the future results will change too much from current version, but it is worth doing and will provide strong support for your statements. Please find my minor comments on specific sections.

Minor comments:

Figure 1a: A sampling gap between KAZ and RUS.

Line 134: What barriers? not clear

Line 142: How did you know the lower differentiation of the Eastern lineage from a phylogenetic tree?

Line 147: Why not HIGH levels of differentiation between the Eastern and Western lineages?

Lines 178-180: Are you sure that western lineage primarily uses croplands during breeding? This partly analysis only shows vegetation differences in ranges of different lineages, does not indicate whether they use or prefer some types of vegetations. I am not sure. Actually, Europe countries obviously have more farming herbaceous than Russia and Africa.

Line 233: Why? How about the remaining non-coding SNPs? You should show your reasons.

Line 242: Showing the values may be more useful for reading.

Lines 281-283: How did you classify SNPs into the four categories? Please clarify.

Line 308: The Eastern ESU showed range contraction, no words for this finding? I feel like you deliberately didn't mention this.

Lines 336-337: See it. The cold adaptation of the Eastern ESU may explain why the eastern ESU underwent range contraction during the past warming phase. If so, you should add the statements.

Line 522: The average read coverage and its range should be provided.

Line 528: Why not use WGS for the 84 individuals? To my knowledge, WGS and ddRAD are priced similarly. BUT, compared to WGS, the amount of data generated by ddRAD is much smaller, and the breadth of coverage of the genome is also much lower. As a result, you only got 73,373 SNPs. Obviously, the limited SNPs number can not provide enough power to support your conclusion as you stated, particularly for the adaptive variations. Perhaps more additional SNPs may be associated with local adaptation, but your data did not cover.

Line 582: What type of data you used to construct the Neighbour-net phylogenetic network? I guess it is the genetic distance matrix, right?

Line 763: There is only one bracket. Some words are missing? I am not sure. Please check it.

Line 779: The divergence time was estimated only based on mitogenomic sequences. I noticed that you used these estimates to match demographic histories that were inferred using WGS data. Why not use assembled gene sequences from WGS data or the concatenated sequence of all SNPs from ddRAD data?

All figures in the main text and SI: Many figure panels are screenshots with low resolution. Please provide clearer vector diagram if this manuscript can be considered.

Reviewer #2

(Remarks to the Author)

I recommend that the manuscript be accepted, as it provides invaluable data for specialists and contributes significantly to the community. The authors present a large set of autosomal and mitochondrial genomes, along with climate data. These data and the associated analyses justify fast publication.

Criticism

The manuscript discusses contrasting responses to climate change in certain species. However, the scientific problem definition is not very clear. The claim that different lineages in different regions have 'contrasting' responses seems intuitively obvious: it is natural for distinct lineages in various regions to respond differently. I am also uncertain if the predicted future dominance of the western and eastern groups can truly be considered 'contrasting.'

Regarding the title, wouldn't it read better as "Evolutionarily..."?

>Predicting species' responses to climate change is a pressing need hampered by our limited >knowledge of spatiotemporal ecological and evolutionary dynamics.

I disagree with this point. Evolution follows its own course. Why is predicting species' responses to climate change so urgent? When you say it's a pressing need, pressing for whom?

>We uncover two evolutionary and ecologically distinct lineages (European and Asian), whose >demographic history, evolutionary divergence, and historical distribution range were profoundly >shaped by past climatic fluctuations.

While past climatic fluctuations undoubtedly shaped these birds, I believe the key question is whether the kestrels exhibit any unexpected proactive adaptations to these ongoing climate changes—adaptations that go beyond natural selection, which is a fundamental aspect of nature.

>Anthropogenic climate change is altering Earth's environmental conditions at an unprecedented rate.

Is this statement entirely accurate? Natural phenomena like meteor strikes and massive volcanic eruptions have caused environmental changes at similarly unprecedented rates. Have glaciers, for example, not caused faster environmental shifts than those driven by human activity?

The western group adapted to agriculture and human expansion, which explains their higher likelihood of persistence. As with sparrows, human expansion can have positive effects on certain species. What specific steps should humans take to aid conservation efforts for such species?

>The two lineages showed very little overlap in their predicted breeding and non-breeding >distributions, reflecting clear differences in lineage-specific climate relationships (Figure 3a; >Figure S12).

Could there be future interbreeding between these lineages, potentially leading to shared SNPs, as the western group expands eastward due to rapid global warming?

In Figure 5, the eastern group was large 130,000 years ago during a period of high temperatures. If Siberian temperatures continue to rise, could this lead to an increase in the eastern group once again?

>Disregarding this information can lead to inaccurate predictions about a species' fate and to >ineffective conservation efforts.

It is difficult to see how this information directly improves conservation strategies.

>Our results revealed that anthropogenic climate change may challenge the future persistence even of >highly mobile taxa, threatening genetic lineages and jeopardizing species' adaptability to >changing ecological conditions.

I am uncertain about this conclusion. Species can adapt not only through genetic changes but also via epigenetic and neuronal (learning) mechanisms, which often precede DNA modification.

Can you clarify how the specimens were actually/practically acquired?

Have you considered competition and interactions with other small raptors, such as common kestrels, as a factor?

Version 1:

Reviewer comments:

Reviewer #1

(Remarks to the Author)

This version of the manuscript has been much improved with addressing all my concerns and providing more details in the Results and Methods sections. I have been convinced by the responses, particular to issues about limited SNP number and WGS data suggestion. The authors have honestly stated that this work was proceeded a few years ago, when the cost of the WGS was higher than the ddRAD-seq. They also did a SNP sampling with different counts for the PCAs, which consistently detect the differentiation between the Western and the Eastern ESUs. This downsampling approach is a great way to prove the credibility and power of ddRAD-seq data. I strongly recommend the authors should add these results and figures into the supplementary materials. I have no additional comments on this version that should be considered for publication after another round of minor revision.

Reviewer #2

(Remarks to the Author)

This reviewer finds the author's point by point arguments demonstrate their clear understanding on their analyses and conclusions. The manuscript is scientifically strong and justified. They have answered, properly, all my questions.

REVIEWER COMMENTS

Reviewer #1 (Remarks to the Author):

Major comments:

This is a very interesting study on predicting lesser kestrels' responses to past climate fluctuations and future climate changes using landscape genomic and demographic approaches and species distribution models based on de novo genome sequencing, whole-genome sequencing, ddRAD sequencing, and climatic data. The authors showed that the genetic divergences and ecological differences of the European and Asian lineages of the lesser kestrel are apparently attributed to past climatic fluctuations, and the cold-adapted Asian lineage may be at higher extinction risk than the warm-adapted European lineage when they face future dramatic climate changes.

>> We thank the reviewer for appreciating the high interest of our research. We also thank the reviewer for carefully assessing our work, which further improved its quality. Please find our point-by-point response below (>> in blue). We have incorporated several changes in the manuscript to address the reviewers' comments, which are highlighted in yellow in the revised manuscript.

Although the authors did lots of analytical works to test their hypothesis and the entire pipeline is quite complete, I am still concerned whether the ddRAD data with limited SNP number (only 73,373 SNPs) and high missing rate (25% individual without genotypes) can provide enough power to interpret genetic differentiation and climatic adaptation. Perhaps, it is enough for genetic differentiation but not for the scans of adaptive SNPs. Additionally, I am not sure whether a large sampling gap between KAZ and RUS populations would affect the population structure and genetic differentiation analyses. I recommend the authors to supplement samples from a few localities in the gap and do the WGS for all individuals. I don't think the future results will change too much from current version, but it is worth doing and will provide strong support for your statements. Please find my minor comments on specific sections.

>> We thank the reviewer for highlighting the thoroughness of our analyses. As there are several points raised here, we will address them one-by-one:

1) Limited SNP number:

Firstly, we would like to highlight that 73k high-quality SNPs is a high number for a ddRAD study. Secondly, we were very careful in how we filtered the dataset. For example, we performed SNP calling using two independent methods (Stacks v2 and BCFtools call) and performed filtering following best practices (Hemstrom et al., 2024). Even with this stringent filtering, we still retained a generous number of markers for our analyses.

2) Missingness in the dataset:

We would like to clarify that a maximum amount of 25% missing data was the threshold used to include a SNP at a per population-level and was not applied globally across the entire dataset (i.e. the $-r$ 0.75 parameter in Stacks, together with the $-p$ 15 parameter). This is a

stringent filtering of our dataset (e.g. see Paris et al., 2017). To address the reviewer's concern, we have calculated the global amount of missing data in the dataset, finding it to be only 1.2%.

3) Power to detect genetic differentiation:

As we are currently unable to generate additional WGS data (see response below to points 6 & 7), we decided to investigate the power of the number of SNPs to detect the main axis of population structure (which differentiates the Western and Eastern ESUs) by downsampling our existing ddRAD dataset.

In the plot above, panels (a) through (f) represent a downsampling of our dataset: (a) the complete linkage-pruned dataset comprising 31,279 SNPs; (b) 20k SNPs; (c) 10k SNPs; (d) 5k SNPs; (e) 2k SNPs; (f) 500 SNPs. Our power to detect differentiation between the Western and the Eastern ESUs remains the same even when downsampling to 2k SNPs. At 500 SNPs, although we start seeing a reduction in the differentiation, PC1 still accounts for 7.23% of the variance and shows some degree of differentiation between the two groups.

4) Power to detect climatic adaptation:

We agree with the reviewer that our approach was somewhat limited in fully addressing the question of climate adaptation. Indeed, it was not our objective to identify the full complement of genetic variation linked to climate adaptation, but rather, to identify some climate-associated genetic variation based on our dataset and assess if this variation differed between the ESUs. We have now modified the sentence at the beginning of the section "Assessing genetic adaptation to local environments" to make this clearer:

"The two ESUs were also differentiated based on climate-associated genetic variation."

We have also included a cautionary note about the use of ddRAD data for identifying adaptive genetic variation in the discussion:

“Whilst these genes represent promising candidates, we appreciate that we have not surveyed the full range of genetic variation associated with climate adaptation, due to, for example, limitations in the ddRAD approach [(Lowry et al. 2017); but see (Catchen et al. 2017); McKinney et al.; 2017], and neglecting the role of structural variation (Mérot et al. 2020).”

However, we want to highlight that our ddRAD approach resulted in an average of 1 ddRAD locus every 80 Kb, which means that most genes were covered by at least one locus. Even if linkage disequilibrium (LD) tends to decay within a few Kb in birds (Spurgin et al., 2024; Secomandi et al., 2023; Bascón-Cardoso et al., 2024), we generally expect LD to extend further than average in regions under selection (Kim & Nielsen, 2004; Nielsen et al., 2005). Thus, it is very possible that our survey (either directly or via linkage), identified a reasonable proportion of the adaptive variation linked to climate. Finally, our careful approach to remove SNPs that were more associated with population structure than with climate (see Supplementary Methods 5) further increases our confidence in the candidate loci we recovered.

5) Sampling gap between KAZ and RUS:

We agree with the reviewer that there is a gap in the genetic sampling between KAZ and RUS. However, we believe this gap is unlikely to affect our conclusions on population structure and genetic differentiation analyses. All our population structure analyses showed that, despite the gap between KAZ (Kazakhstan) and RUS (Russia), KAZ clusters with RUS, MON (North Mongolia) and MOS (South Mongolia) in the Eastern ESU. If population differentiation in the lesser kestrel was consistent with a scenario of isolation-by-distance (IBD), we would expect KAZ to cluster with TUR (Turkey) and ISR (Israel) given the lower geographic distance between KAZ and these localities (TUR-KAZ: 1435 km), ISR-KAZ: 1997 km) compared to the geographic distance between KAZ and RUS (KAZ-RUS: 3384 km). Indeed, this is the reason why we see a main barrier to gene flow between KAZ and the Western ESU localities in the EEMS analysis coinciding with the Caucasus mountains (Figure 2c).

6) Supplementing samples:

Unfortunately, despite our efforts to build a broad scientific collaboration including dozens of colleagues from 13 countries, this sampling gap could not be filled. This was due to the fact that we were unable to find any local scientists from this region that were able to collect samples, and our funding was insufficient to organise dedicated scientific expeditions.

7) WGS:

Firstly, whilst we recognise the benefits of WGS over ddRAD, this research project began six years ago when the price difference between WGS and ddRAD was still significant. In 2018, it would have cost ~200 € per sample for WGS compared to 35 € per sample to perform ddRAD. Unfortunately, the high price for WGS meant it was near impossible for us to sequence all samples using this technique with the available funding.

Overall, we strongly believe that our dataset was adequate to address our main objectives, as highlighted at the end of the introduction:

“Provide an in-depth assessment of the vulnerability of the lesser kestrel to climate change by (1) inferring intraspecific evolutionary lineages and assessing their ecological differentiation, and (2) investigating lineage-specific genomic, demographic and distributional responses to past, present and future climatic fluctuations across its global distribution range.”

We hope this response will convince the reviewer and the editor that the employment of WGS was unfeasible for this project and that its use would not have changed our main objectives, nor outcomes.

Minor comments:

Figure 1a: A sampling gap between KAZ and RUS.

>> Please see our response to the major comment above.

Line 134: What barriers? not clear

>> We thank the reviewer for highlighting this omission. We realise that we had not specified to which analyses this statement made reference to and have now amended the sentence to include clearer information:

“Using the Estimated Effective Migration Surfaces (EEMS) method, based on a stepping-stone dispersal model, we detected a main barrier to gene flow between the Western and Eastern lineages coinciding with the Caucasus mountains...”

Line 142: How did you know the lower differentiation of the Eastern lineage from a phylogenetic tree?

>> Thank you for this comment. We originally wrote this section under the premise that in Treemix, horizontal branch lengths are proportional to the amount of genetic drift that has occurred on the branch (*i.e.* drift parameter). Given the lower drift parameter values for Eastern populations compared to Western populations, we expect lower differentiation between the outgroup and the Eastern lineage. We have thus modified this sentence to focus on the lower genetic distances between the Eastern lineage and the outgroup as shown in the Neighbour-net network:

“The combination of 1) lower genetic distances between the Eastern lineage and the outgroup compared to the Western lineage (as shown in the neighbour-net network), and 2) the outgroup branch rooting the network within the Eastern lineage (Supplementary Fig. 7), point to an Asian origin for the species.”

Line 147: Why not HIGH levels of differentiation between the Eastern and Western lineages?

>> We believe the reviewer may have misinterpreted this sentence, and this is likely due to it not being very clear previously. With this sentence, we wished to stress that despite the differentiation between the Western and the Eastern lineages being the main axis of population structure in the data, their genetic differentiation is still quite low. One of the reasons could be the fact that there is still some gene flow between the lineages. We have now made this sentence clearer and added quantitative values of genetic differentiation:

“We also inferred substantial contemporary gene flow among populations in the Western lineage, as well as some gene flow from the Eastern to the Western lineage (Fig. 2c), the latter consistent with low levels of genetic differentiation between the two lineages ($\Phi_{ST} = 0.03 - 0.05$; Supplementary Fig. 2)..

Lines 178-180: Are you sure that western lineage primarily uses croplands during breeding? This partly analysis only shows vegetation differences in ranges of different lineages, does not indicate whether they use or prefer some types of vegetations. I am not sure. Actually, Europe countries obviously have more farming herbaceous than Russia and Africa.

>> We thank the reviewer for this comment. Yes, it is well known that lesser kestrels from the western lineage primarily use (and show positive selection for) croplands both for colony settlement and as foraging habitats during breeding, especially during nestling rearing (e.g. Tella et al. 1998; Catry et al. 2012; Morganti et al. 2021; Assandri et al. 2023). Our findings that breeding sites are mostly (and preferentially) located in cropland is therefore not surprising and entirely in line with this previous body of evidence. Our analyses assessed habitat use based on occurrence records and compared it with the availability within a buffer of 1500 km surrounding each occurrence record, unequivocally showing that croplands were preferred during breeding. The same analysis applied to the eastern lineage revealed opposite trends, showing avoidance of cropland and a preference for natural open habitats (steppes). Hence, we not only show that the two lineages settle in different habitats, but that they also differ in the extent to which they prefer those habitats, at least during breeding.

Line 233: Why? How about the remaining non-coding SNPs? You should show your reasons.

>> We have now included a sentence to explain the reasons why we only selected climate-associated SNPs within protein-coding genes:

“We focused our analyses on SNPs within protein-coding genes because these are more likely to be in linkage with genetic variation potentially associated with climate adaptation.”

Line 242: Showing the values may be more useful for reading.

>> As we understand, the reviewer is referring specifically to the turnover values. Turnover values themselves are dependent on the SNPs used and are therefore not that useful for interpretation. However, to address this point, we have decided to make explicit reference to the element of Figure 4c that readers should focus on to interpret the comparison of the turnover functions for climate-associated SNPs vs. reference SNPs:

“A gradient forest (GF) analysis applied to the 61 climate-associated SNPs showed that they were more strongly associated with climatic variables than neutral SNPs, as highlighted by the higher turnover values based on the top-ranked bioclimatic variables (BIO19 and BIO7; Supplementary Fig. 13, Fig. 4c; thick blue and red lines compared to thick black lines).”

Lines 281-283: How did you classify SNPs into the four categories? Please clarify.

>> We agree with the reviewer that this section required further clarification in the legend of Figure 4. We have now mentioned that the involvement of these genes in these processes was determined through a literature review:

“(Right) Tiles are coloured in grey when the candidate SNP is found within a gene identified, through a literature search (see Methods), as being associated with local adaptation (first column), phenotypic traits important for local adaptation (second column), adaptation to urban environments or domestication (third column), and/or stress response (fourth column) in vertebrates.”

We wish to highlight that these categories were previously explained in full detail in the Methods section:

“To explore the potential roles of genes with climate-associated SNPs in local climate adaptation, we conducted literature searches in Web of Science (www.webofscience.com) and Google Scholar (www.scholar.google.com) with the keywords: (1) “local adaptation”, (2) “phenotypic traits”, (3) “domestication/urban adaptation” and (4) “stress response”. We selected articles that focused on vertebrate species and that reported associations between the candidate genes and the keywords.”

Line 308: The Eastern ESU showed range contraction, no words for this finding? I feel like you deliberately didn't mention this.

>> As the reviewer mentions in the next comment, this range contraction was already discussed in the next section of our manuscript.

To be clear, the section: “Effects of climatic fluctuations on past demographic history and changes in distribution range” focuses on explaining the demographic and distribution results between the Last Interglacial and the beginning of the Holocene.

The next section, “Distributional and genetic responses to future climate” focuses on forecasted trends under future climate and how they are consistent with trends throughout the Holocene.

Lines 336-337: See it. The cold adaptation of the Eastern ESU may explain why the eastern ESU underwent range contraction during the past warming phase. If so, you should add the statements.

>> Thank you for noticing this. We agree that these considerations were not made explicit in the text and we have added them for clarity:

“Under extreme warming future climate, the breeding and non-breeding ranges are predicted to expand substantially for the warmer-adapted Western ESU and are predicted to shrink for the colder-adapted Eastern ESU (Fig. 6a,b), consistent with their adaptations to different climates. Indeed, these predicted changes are consistent with hindcasted trends in range size through the Holocene, although they show a greater magnitude of change compared to the latter (Fig. 6b).”

Line 522: The average read coverage and its range should be provided.

>> We agree that this is important information that should be added. The average and range of per-sample depth of coverage have now been included in the methods:

“The average per-sample depth of coverage was 45x (min-max: 16-68x).”

Line 528: Why not use WGS for the 84 individuals? To my knowledge, WGS and ddRAD are priced similarly. BUT, compared to WGS, the amount of data generated by ddRAD is much smaller, and the breadth of coverage of the genome is also much lower. As a result, you only got 73,373 SNPs. Obviously, the limited SNPs number can not provide enough power to support your conclusion as you stated, particularly for the adaptative variations. Perhaps more additional SNPs may be associated with local adaptation, but your data did not cover.

>> Please see our response to the major comment above.

Line 582: What type of data you used to construct the Neighbour-net phylogenetic network? I guess it is the genetic distance matrix, right?

>> Yes it was a genetic distance matrix. We have now included this information in the manuscript:

“We also inferred a Neighbour-net phylogenetic network (Bryant & Moulton, 2004) using a genetic distance matrix, implemented in SplitsTree5 v.5.0.16 (Huson & Bryant, 2006).”

More information on this analysis about it can be found in the Supplementary Information:

“To further visualise genealogical patterns, we inferred a Neighbour-net phylogenetic network (Bryant & Moulton, 2004), implemented in SplitsTree5 v.5.0.16 (Huson & Bryant, 2006). We first converted the VCF file to a DNABin object using the vcfR2DNABin function from the vcfR R package (Knaus & Grünwald, 2017) and then computed a genetic distance matrix using the dist.dna function from the ape R package (Paradis & Schliep, 2019), which we used as input for SplitsTree5.”

Line 763: There is only one bracket. Some words are missing? I am not sure. Please check it.

>> Thank you for noticing this. We have now added the missing bracket:

“The latter were selected based on their overall importance in the GF analysis, starting from

the highest ranking variable and keeping those variables showing limited collinearity with previously included ones (*i.e.*, $|r| < 0.7$).”

Line 779: The divergence time was estimated only based on mitogenomic sequences. I noticed that you used these estimates to match demographic histories that were inferred using WGS data. Why not use assembled gene sequences from WGS data or the concatenated sequence of all SNPs from ddRAD data?

>> We thank the reviewer for this interesting point. Actually, the divergence time was estimated both using mitogenomes and through demographic modelling in DIYABC using the ddRAD data. These estimates were consistent as it is reported in the results:

“The split time between Western and Eastern ESUs was estimated at 40.4 kya (95% CI: 15.8-68.7 kya), during the second half of the Last Glacial Period (115-11.7 kya), coinciding with a steady demographic decline inferred by the demographic analysis (Fig. 5c) and a decrease in the extent of breeding range (Fig. 5e,g). The divergence time between the mitochondrial haplogroups A (most frequent in the Western ESU) and B (most frequent in the Eastern ESU) was inferred at 49 kya (41.7-56.3 kya) (Supplementary Fig. 14), corroborating the evidence that the split between the two ESUs took place during the Last Glacial Period.”

We opted for an approach using demographic modelling for ddRAD data rather than a phylogenetic dating approach. Phylogenetic dating using concatenation-based methods is known to overestimate divergence times mostly because they fail to fully consider the role of incomplete lineage sorting (Angelis & Dos Reis, 2015), particularly at shallow evolutionary timescales. This can be partly mitigated through the use of multispecies coalescent methods (*e.g.* Stange et al. 2018), but these models assume complete isolation after species divergence, with no migration, hybridization, or introgression. Given that we detected contemporary gene flow among lesser kestrel populations, we decided to use a method where gene flow could be explicitly accounted for when estimating divergence times.

All figures in the main text and SI: Many figure panels are screenshots with low resolution. Please provide clearer vector diagram if this manuscript can be considered.

>> We apologise for the inconvenience that the low resolution of some figure panels might have caused during the review process. We believe this might have been caused by the reduction in resolution when uploading the documents on the submission portal. We have now uploaded high-quality vector images.

Reviewer #2 (Remarks to the Author):

I recommend that the manuscript be accepted, as it provides invaluable data for specialists and contributes significantly to the community. The authors present a large set of autosomal and mitochondrial genomes, along with climate data. These data and the associated analyses justify fast publication.

>> We thank Reviewer #2 for their positive feedback on our manuscript, and for appreciating the interest of our results to the scientific community and the scale of our dataset. We also

thank Reviewer #2 for their insightful comments that have strengthened the manuscript. Please find our point-by-point responses below.

Criticism

The manuscript discusses contrasting responses to climate change in certain species. However, the scientific problem definition is not very clear. The claim that different lineages in different regions have 'contrasting' responses seems intuitively obvious: it is natural for distinct lineages in various regions to respond differently. I am also uncertain if the predicted future dominance of the western and eastern groups can truly be considered 'contrasting.'

>> We agree with the reviewer that responses to climate change might be expected to differ between different lineages and in different regions. However, surprisingly, most climate change vulnerability assessments have ignored potential differences in vulnerability among populations within species (see Razgour et al., 2019). We wish to emphasise that our study provides a thorough investigation into several ecological and evolutionary factors determining the drivers of different responses to climate change among lineages within a species.

Given the first comment from Reviewer 2, we understand that the reviewer believes the presented research is of high scientific value. It is therefore our understanding that the reviewer does not favour our use of the word "contrasting" to describe our findings. We have considered alternatives that may be more suitable, and we have changed "contrasting" to "divergent" to describe the different trends we observe between the two lineages, both in the title, and throughout the manuscript.

Regarding the title, wouldn't it read better as "Evolutionarily..."?

>> Thank you for suggesting this change. We agree with using the word Evolutionarily and have changed the title accordingly.

>Predicting species' responses to climate change is a pressing need hampered by our limited >knowledge of spatiotemporal ecological and evolutionary dynamics.

I disagree with this point. Evolution follows its own course. Why is predicting species' responses to climate change so urgent? When you say it's a pressing need, pressing for whom?

>> Thank you for this comment. We reflected on the tone of this sentence and we agreed that we were "putting the cart before the horse". We have now removed "pressing need" from the sentence:

"Accurately predicting species' responses to anthropogenic climate change is hampered by limited knowledge of their spatiotemporal ecological and evolutionary dynamics."

>We uncover two evolutionary and ecologically distinct lineages (European and Asian), whose >demographic history, evolutionary divergence, and historical distribution range were profoundly >shaped by past climatic fluctuations.

While past climatic fluctuations undoubtedly shaped these birds, I believe the key question is whether the kestrels exhibit any unexpected proactive adaptations to these ongoing climate changes—adaptations that go beyond natural selection, which is a fundamental aspect of nature.

>> These are thoughtful and nuanced points. We can consider the following:

- Behavioural flexibility - e.g. altering the timing of migration and reproduction or range shifts in response to climate.
- Plasticity or resilience - i.e. the ability of individuals within the populations to adjust to new climatic conditions without requiring genetic changes.

We have now included these points in the discussion:

“However, considering the apparent range flexibility inferred under past climate change, populations might relocate further north, where climatic conditions are expected to become progressively more suitable in the near future. Phenological adjustments (*i.e.*, advances) in the timing of reproduction to avoid breeding with extreme climatic conditions (which may heavily decrease fitness, e.g. late spring/early summer heatwaves which cause extensive nestling mortality; Catry et al. 2011, Corregidor-Castro et al., 2023), along with the potential for phenotypic plasticity (Corregidor-Castro et al., 2024), represent promising avenues for adaptation to future climate.”

And also in our conclusion paragraph:

“The resilience of highly mobile species to a rapidly changing climate will ultimately depend on their ability to adaptively respond to novel conditions through different mechanisms, including dispersal, phenotypic plasticity and genetic adaptation.”

Specifically, we think that including this in the conclusion of the manuscript aligns well with our introduction:

“To track fast-changing environments, organisms can relocate to more suitable areas, respond via phenotypic plasticity, or undergo rapid adaptation”.

Overall, these additions make the present research more relevant for understanding how the species might navigate ongoing and future climate challenges.

>Anthropogenic climate change is altering Earth’s environmental conditions at an unprecedented rate.

Is this statement entirely accurate? Natural phenomena like meteor strikes and massive volcanic eruptions have caused environmental changes at similarly unprecedented rates. Have glaciers, for example, not caused faster environmental shifts than those driven by human activity?

>> We agree that the phrasing was previously too dramatic. We have amended this sentence to read:

“Anthropogenic climate change is rapidly altering Earth’s environmental conditions, exacerbating the effects of land use change ...”

The western group adapted to agriculture and human expansion, which explains their higher likelihood of persistence. As with sparrows, human expansion can have positive effects on certain species. What specific steps should humans take to aid conservation efforts for such species?

>> Again, a thought-provoking comment. Although the western lineage has clearly greatly benefited from the spread of agriculture and land transformation in the past millennia, similarly to many farmland bird species, the reviewer’s statement pertaining to a higher persistence in relation to adapting to agriculture and human expansion is not necessarily true. Agricultural intensification following the 20th century so-called “Green Revolution”, with increasing use of pesticides, the establishment of monocultures, and maximisation of crop yields, has profoundly modified croplands worldwide, making these habitats much less suitable for several species that had previously been thriving in agroecosystems for many centuries (Batáry et al., 2020). Indeed, farmland birds are declining worldwide more than any other group of birds (Stanton et al., 2018; Rigal et al., 2023). Amongst the literature on farmland bird species, there are plenty of studies suggesting which steps should be adopted to promote their conservation in farmland habitats. In fact, some of these (favouring crop rotation, incentivising forage crops and crop diversification, and reducing insecticide use) apply to lesser kestrels (see e.g. Wilson et al. 2009). Still, we feel it would be out of the scope of our study to provide such suggestions in the context of the present study.

>The two lineages showed very little overlap in their predicted breeding and non-breeding >distributions, reflecting clear differences in lineage-specific climate relationships (Figure 3a; >Figure S12).

Could there be future interbreeding between these lineages, potentially leading to shared SNPs, as the western group expands eastward due to rapid global warming?

>> Yes, we fully agree and indeed have addressed this:

“Since the Western ESU is currently adapted to warmer environments than the Eastern ESU, the introduction of adaptive variation via gene flow from the former to the latter might make populations more resistant to warmer future climates and avoid local extirpation”

In Figure 5, the eastern group was large 130,000 years ago during a period of high temperatures. If Siberian temperatures continue to rise, could this lead to an increase in the eastern group once again?

>> Thank you for pointing this out. To be clear, there was no increase in the demographic size of the eastern lineage during this time. Instead, there was a larger availability of habitat as shown by the SDMs presented in Figure 5g. The past projection of the SDMs is not based solely on temperature, but instead includes four variables: 1) annual mean temperature, 2)

temperature annual range, 3) annual precipitation, and 4) precipitation seasonality. Therefore, we cannot conclude that a rise in temperature observed 130,000 years ago was the main driver of this area of increased habitat suitability.

>Disregarding this information can lead to inaccurate predictions about a species' fate and to >ineffective conservation efforts.

It is difficult to see how this information directly improves conservation strategies.

>> We politely disagree with the reviewer on this point. There is a large body of literature highlighting the need for conservation strategies to incorporate intraspecific diversity and ESUs from which some examples are given in this section of the introduction (*i.e.* Hällfors et al., 2016; Razgour et al., 2019). In addition, we argue that understanding species and populations' demographic sensitivity to past climatic fluctuations can also be key for planning conservation efforts as it allows the identification of species prone to demographic decline under current and future climate changes (*e.g.* Germain et al. 2023). The anticipation of these responses enables proactive conservation planning, including improving the quality of foraging/nesting habitats in those areas that are deemed to become climatically more suitable in the near future, and strengthening local populations at range margins in areas that are going to become highly suitable, perhaps even by performing translocations. There are several examples of these kinds of actions in recently funded/completed LIFE projects targeting the lesser kestrel.

>Our results revealed that anthropogenic climate change may challenge the future persistence even of >highly mobile taxa, threatening genetic lineages and jeopardizing species' adaptability to >changing ecological conditions.

I am uncertain about this conclusion. Species can adapt not only through genetic changes but also via epigenetic and neuronal (learning) mechanisms, which often precede DNA modification.

>> We agree that we had ended the conclusions without mention of potential mechanisms through which highly mobile taxa could adapt to their changing environments. We have now included additional text pertaining to this in the conclusions:

“Yet, our results revealed that anthropogenic climate change may challenge the future persistence even of highly mobile taxa, threatening genetic lineages and jeopardizing species' adaptability to changing ecological conditions. The resilience of highly mobile species in a changing planet will depend on their ability to relocate to more suitable habitats, adjust their phenology, and adapt through rapid genetic changes and phenotypic plasticity.”

Can you clarify how the specimens were actually/practically acquired?

>> We did not collect any specimens for this study. For genetic analyses, blood samples were collected directly from the sampled individuals, as described in the methods:

“We collected blood (approx. 50 µl) from 119 unrelated nestlings and breeding adults at 16 localities spread across the whole breeding range”

Details of collection permits are reported in the Acknowledgements.

Have you considered competition and interactions with other small raptors, such as common kestrels, as a factor?

>> We thank the reviewer for this interesting comment. We did not consider interspecific interactions in this manuscript, as it would be complicated to model these effects (e.g. in our species' distribution models or in habitat suitability analyses) due to the lack of detailed distributional data and the limited knowledge about the importance of these interactions in shaping distributions. Yet, we note that at the global breeding range scale, there is a huge overlap between lesser kestrels and two other ecologically and phylogenetically related species, the common kestrel and the red-footed falcon, suggesting that interspecific interactions do not play a key role in shaping their large-scale distribution. Moreover, we have recently investigated niche overlap between these three species in a portion of their distribution range where they co-occur syntopically (Po Plain, northern Italy) (Berlusconi et al., 2022). Even if some negative interactions have been observed between red-footed falcons and lesser kestrels (with red-footed falcons kleptoparasitizing lesser kestrels, Berlusconi et al., 2024), they do not seem to prevent these species from co-occurring at the same sites, consistent with their large range overlap across Eurasia. We briefly acknowledge these points in the Methods section:

“We relied on species distribution models (SDMs) to reconstruct the current breeding and non-breeding distribution of lesser kestrels, and used these models to infer both past and future distribution, assuming consistent lineage-specific climate relationships across time and no effects of interspecific competition. The latter assumption is based on the fact that lesser kestrels largely co-occur with other ecologically and phylogenetically related species (e.g. *F. vespertinus*, *F. tinnunculus*) across their breeding distribution range (Cramp, 1998) and often occur syntopically (Berlusconi et al., 2022).”

>> **References**

Angelis, K., & Dos Reis, M. (2015). The impact of ancestral population size and incomplete lineage sorting on Bayesian estimation of species divergence times. *Current Zoology*, 61(5), 874–885. <https://doi.org/10.1093/czoolo/61.5.874>

Bascón-Cardozo, K., Bours, A., Manthey, G., Durieux, G., Dutheil, J. Y., Pruißcher, P., ... & Liedvogel, M. (2024). Fine-Scale Map Reveals Highly Variable Recombination Rates Associated with Genomic Features in the Eurasian Blackcap. *Genome Biology and Evolution*, 16(1), evad233.

Batáry, P., Báldi, A., Ekroos, J., Gallé, R., Grass, I., & Tschardtke, T. (2020). *Biologia Futura: landscape perspectives on farmland biodiversity conservation*. *Biologia Futura*, 71(1), 9-18.

Berlusconi, A., Preatoni, D., Assandri, G., Bisi, F., Brambilla, M., Cecere, J. G., ... & Morganti, M. (2022). Intra-guild spatial niche overlap among three small falcon species in an area of recent sympatry. *The European Zoological Journal*, 89(1), 510-526.

Berlusconi, A., Scridel, D., Eberle, L., Martinoli, A., Bazzi, G., Assandri, G., ... & Morganti, M. (2024). Ecological and social factors affecting the occurrence of kleptoparasitism in two recently established sympatric breeding falcons. *Behavioral Ecology and Sociobiology*, 78(2), 14.

Burns, F., Eaton, M. A., Burfield, I. J., Klvaňová, A., Šilarová, E., Staneva, A., & Gregory, R. D. (2021). Abundance decline in the avifauna of the European Union reveals cross-continental similarities in biodiversity change. *Ecology and Evolution*, 11(23), 16647-16660.

Bryant, D., & Moulton, V. (2004). Neighbor-net: an agglomerative method for the construction of phylogenetic networks. *Molecular Biology and Evolution*, 21(2), 255–265.

Catchen, J. M., Hohenlohe, P. A., Bernatchez, L., Funk, W. C., Andrews, K. R., & Allendorf, F. W. (2017). Unbroken: RADseq remains a powerful tool for understanding the genetics of adaptation in natural populations. *Molecular ecology resources*, 17(3), 362-365.

Corregidor-Castro, A., Romano, A., Butler, M., Cecere, J. G., Morinay, J., Morganti, M., ... & Pilastro, A. (2024). Temperature-related developmental plasticity, not selection, affects offspring body size and shape in a bird of prey. *EcoEvoRxiv*.
<https://doi.org/10.32942/X2G04G>

Dormann, C. F., Elith, J., Bacher, S., Buchmann, C., Carl, G., Carré, G., ... & Lautenbach, S. (2013). Collinearity: a review of methods to deal with it and a simulation study evaluating their performance. *Ecography*, 36(1), 27–46.

Germain, R. R., Feng, S., Chen, G., Graves, G. R., Tobias, J. A., Rahbek, C., Lei, F., Fjeldså, J., Hosner, P. A., Gilbert, M. T. P., Zhang, G., & Nogués-Bravo, D. (2023). Species-specific traits mediate avian demographic responses under past climate change. *Nature Ecology & Evolution*, 7(6), 862–872.

Hällfors, M. H., Liao, J., Dzurisin, J., Grundel, R., Hyvärinen, M., Towle, K., Wu, G. C., & Hellmann, J. J. (2016). Addressing potential local adaptation in species distribution models: implications for conservation under climate change. *Ecological Applications: A Publication of the Ecological Society of America*, 26(4), 1154–1169.

Hemstrom, W., Grummer, J. A., Luikart, G., & Christie, M. R. (2024). Next-generation data filtering in the genomics era. *Nature Reviews Genetics*, 1-18.

Hoban, S., Hvilson, C., Aissi, A., Aleixo, A., Bélanger, J., Biala, K., ... & da Silva, J. M. (2024). How can biodiversity strategy and action plans incorporate genetic diversity and align with global commitments?. *BioScience*, biae106.

Huson, D. H., & Bryant, D. (2006). Application of phylogenetic networks in evolutionary studies. *Molecular Biology and Evolution*, 23(2), 254–267.

Kim, Y., & Nielsen, R. (2004). Linkage disequilibrium as a signature of selective sweeps. *Genetics*, 167(3), 1513-1524.

Knaus, B. J., & Grünwald, N. J. (2017). vcfr: a package to manipulate and visualize variant call format data in R. *Molecular Ecology Resources*, 17(1), 44–53.

Nielsen, R., Williamson, S., Kim, Y., Hubisz, M. J., Clark, A. G., & Bustamante, C. (2005). Genomic scans for selective sweeps using SNP data. *Genome research*, 15(11), 1566-1575.

Paradis, E., & Schliep, K. (2019). ape 5.0: an environment for modern phylogenetics and evolutionary analyses in R. *Bioinformatics*, 35(3), 526–528.

Paris, J. R., Stevens, J. R., & Catchen, J. M. (2017). Lost in parameter space: a road map for stacks. *Methods in Ecology and Evolution*, 8(10), 1360-1373.

Petkova, D., Novembre, J., & Stephens, M. (2016). Visualizing spatial population structure with estimated effective migration surfaces. *Nature genetics*, 48(1), 94-100.

Razgour, O., Forester, B., Taggart, J. B., Bekaert, M., Juste, J., Ibáñez, C., Puechmaille, S. J., Novella-Fernandez, R., Alberdi, A., & Manel, S. (2019). Considering adaptive genetic variation in climate change vulnerability assessment reduces species range loss projections. *Proceedings of the National Academy of Sciences of the United States of America*, 116(21), 10418–10423. <https://doi.org/10.1073/pn>

Rigal, S., Dakos, V., Alonso, H., Auniš, A., Benkő, Z., Brotons, L., ... & Devictor, V. (2023). Farmland practices are driving bird population decline across Europe. *Proceedings of the National Academy of Sciences*, 120(21), e2216573120.

Secomandi, S., Gallo, G. R., Sozzoni, M., Iannucci, A., Galati, E., Abueg, L., ... & Formenti, G. (2023). A chromosome-level reference genome and pangenome for barn swallow population genomics. *Cell reports*, 42(1).

Spurgin, L. G., Bosse, M., Adriaensen, F., Albayrak, T., Barboutis, C., Belda, E., ... & Slate, J. (2024). The great tit HapMap project: A continental- scale analysis of genomic variation in a songbird. *Molecular ecology resources*, 24(5), e13969.

Stange, M., Sánchez-Villagra, M. R., Salzburger, W., & Matschiner, M. (2018). Bayesian divergence-time estimation with genome-wide single- nucleotide polymorphism data of sea catfishes (Ariidae) supports miocene closure of the panamanian isthmus. *Systematic Biology*, 67(4), 681–699. <https://doi.org/10.1093/sysbio/syy006>

Stanton, R. L., Morrissey, C. A., & Clark, R. G. (2018). Analysis of trends and agricultural drivers of farmland bird declines in North America: A review. *Agriculture, Ecosystems & Environment*, 254, 244-254.

Wilson, J. D., Evans, A. D., & Grice, P. V. (2009). *Bird conservation and agriculture*. Cambridge University Press.

REVIEWERS' COMMENTS

Reviewer #1 (Remarks to the Author):

This version of the manuscript has been much improved with addressing all my concerns and providing more details in the Results and Methods sections. I have been convinced by the responses, particular to issues about limited SNP number and WGS data suggestion. The authors have honestly stated that this work was proceeded a few years ago, when the cost of the WGS was higher than the ddRAD-seq. They also did a SNP sampling with different counts for the PCAs, which consistently detect the differentiation between the Western and the Eastern ESUs. This downsampling approach is a great way to prove the credibility and power of ddRAD-seq data. I strongly recommend the authors should add these results and figures into the supplementary materials. I have no additional comments on this version that should be considered for publication after another round of minor revision.

>> We thank Reviewer #1 for their comments on our manuscript. We have followed their advice and included the results and the figure (Supplementary Figure 3) related to the downsampling approach in the Supplementary information.

Reviewer #2 (Remarks to the Author):

This reviewer finds the author's point by point arguments demonstrate their clear understanding on their analyses and conclusions. The manuscript is scientifically strong and justified. They have answered, properly, all my questions.

>> We thank Reviewer #2 for reviewing our manuscript and thank them for their positive comments.